# GUIDANCE MATTERS: RETHINKING THE EVALUATION PITFALL FOR TEXT-TO-IMAGE GENERATION

**Dian Xie**[1]  **Shitong Shao**[1]  **Lichen Bai**[1]  **Zikai Zhou**[1]
**Bojun Cheng**[1]  **Shuo Yang**[2]  **Jun Wu**[3]  **Zeke Xie**[1]†
[1]The Hong Kong University of Science and Technology (Guangzhou)
[2]Harbin Institute of Technology (Shenzhen)
[3]Cogniser Information Technology
`{dxie810, sshao213, lbai292, zzhou440}@connect.hkust-gz.edu.cn`
`zekexie@hkust-gz.edu.cn`

## ABSTRACT

Classifier-free guidance (CFG) has helped diffusion models achieve great conditional generation in various fields. Recently, more diffusion guidance methods have emerged with improved generation quality and human preference. However, can these emerging diffusion guidance methods really achieve solid and significant improvements? In this paper, we rethink recent progress on diffusion guidance. Our work mainly consists of four contributions. First, we reveal a critical evaluation pitfall that common human preference models exhibit a strong bias towards large guidance scales. Simply increasing the CFG scale can easily improve quantitative evaluation scores due to strong semantic alignment, even if image quality is severely damaged (e.g., oversaturation and artifacts). Second, we introduce a novel guidance-aware evaluation (GA-Eval) framework that employs effective guidance scale calibration to enable fair comparison between current guidance methods and CFG by identifying the effects orthogonal and parallel to CFG effects. Third, motivated by the evaluation pitfall, we design Transcendent Diffusion Guidance (TDG) method that can significantly improve human preference scores in the conventional evaluation framework but actually does not work in practice. Fourth, in extensive experiments, we empirically evaluate recent eight diffusion guidance methods within the conventional evaluation framework and the proposed GA-Eval framework. Notably, simply increasing the CFG scales can compete with most studied diffusion guidance methods, while all methods suffer severely from winning rate degradation over standard CFG. Our work would strongly motivate the community to rethink the evaluation paradigm and future directions of this field. Our code is available at https://github.com/Maxwells-Demons/Guidance-Matters.

## 1 INTRODUCTION

In recent years, diffusion models (Sohl-Dickstein et al., 2015; Ho et al., 2020; Song et al., 2021a;b; Lipman et al., 2023) have become cutting-edge techniques for generating high-quality contents, including image (Nichol et al., 2022; Rombach et al., 2022; Peebles & Xie, 2023; Esser et al., 2024), video (Ho et al., 2022; Blattmann et al., 2023b;a; Shao et al., 2025b), and 3D assets (Poole et al., 2023; Yi et al., 2024). The flexibility of diffusion process enabled versatile inference and training process (Shao et al., 2024; 2025c; Zhou et al., 2025; Huang et al., 2025), and can be applied to tasks in other domains (Li et al., 2025). A keys to empirical success lies on the powerful conditional and controllable generation ability of diffusion models, empowering rich AIGC applications.

Classifier-free guidance (CFG) (Ho & Salimans, 2021) is exactly a very popular and powerful guidance method that enables direct control over the denoising process with text prompts. Recent studies (Hong et al., 2023; Hong, 2024; Ahn et al., 2024; Si et al., 2024; Chung et al., 2025; Bai et al.,

---

† Corresponding author.

Prompt: *"An astronaut riding a horse."*

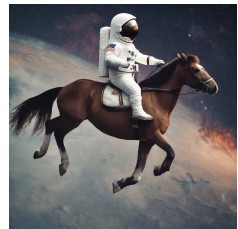 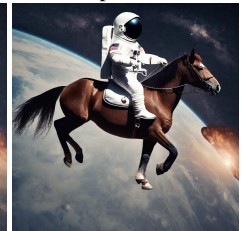 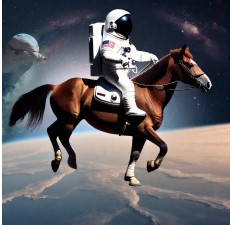 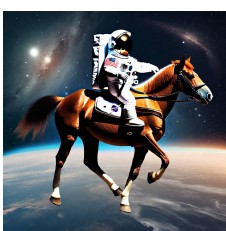

HPS v2: 31.4546     HPS v2: 32.2331     HPS v2: 32.5681     HPS v2: 32.5991

Figure 1: The HPS v2 scores of generated images under different CFG scales $\omega \in \{5.5, 10, 15, 20\}$, respectively. Model: Stable Diffusion-XL. HPSv2 exhibit a strong bias to large CFG scales in a wide range, even if generation quality starts to degrade due to too strong guidance.

2025a; Shao et al., 2025a; Zhou et al., 2025) have proposed several diffusion sampling or guidance methods to improve conditional generation quality. Meanwhile, in those studies, the evaluation of text-to-image generation starts to mainly depend on recent human preference models, such as HPS v2 (Wu et al., 2023), PickScore (Kirstain et al., 2023), and ImageReward (Xu et al., 2024), which are considered to be superior to conventional evaluation metrics. This is commonly true in human preference datasets.

However, researcher overlooked that human often prefer images with gorgeous colors (Lin et al., 2024), which are correlated to highly saturated images generated with a large guidance scale. This overlooked phenomenon has led to a serious evaluation pitfall that human preference models would give better ratings on images generated with a relatively large guidance scale, as illustrated in Fig 1. We present several qualitative cases in Fig 2. It is easy for CFG to achieve comparable or better performance to many methods by simply increasing the guidance scale.

We raise a key question: may recent advanced diffusion guidance methods overfit the overlooked large-guidance bias in human preference models? In this work, to validate how much the recent advanced methods' performance is biased by the evaluation pitfall, we try to disentangle their effects from large CFG effects by calculating the effective guidance scale and then comparing the winning rates between a method and its effective CFG scale. By evaluating these methods with CFG under the same effective guidance scales, we can avoid such overlooked evaluation pitfall. We surprisingly discovered that most methods mainly exploited large guidance scale to achieve high performance, as their winning rates dropped considerably compared with effective CFG.

Our contributions can be summarized as follows:

- First, we reveal a critical evaluation pitfall that common human preference models, such as HPSv2 and ImageReward, exhibit a strong bias towards large guidance scales. As Fig 1 shows, simply increasing the CFG scale can easily improve quantitative evaluation scores due to strong semantic alignment, even if image quality is severely damaged (e.g., oversaturation and artifacts).

- Second, we introduce a novel guidance-aware evaluation (GA-Eval) framework that employs effective guidance scale calibration to enable fair comparison between current guidance methods and CFG by identifying the effects orthogonal and parallel to CFG effects.

- Third, motivated by the evaluation pitfall, we easily design Transcendent Diffusion Guidance (TDG) method that imitate the creation of weak condition in other methods during the sampling process. This is exactly an example method that can significantly improve human preference scores in the conventional evaluation framework but cannot actually improve generation quality over CFG.

- Fourth, in extensive experiments, we empirically evaluated eight diffusion guidance methods within the conventional evaluation framework and the proposed GA-Eval framework. Notably, simply increasing the CFG scales can compete with most studied diffusion guidance methods, while all methods suffer severely from winning rate degradation over CFG.

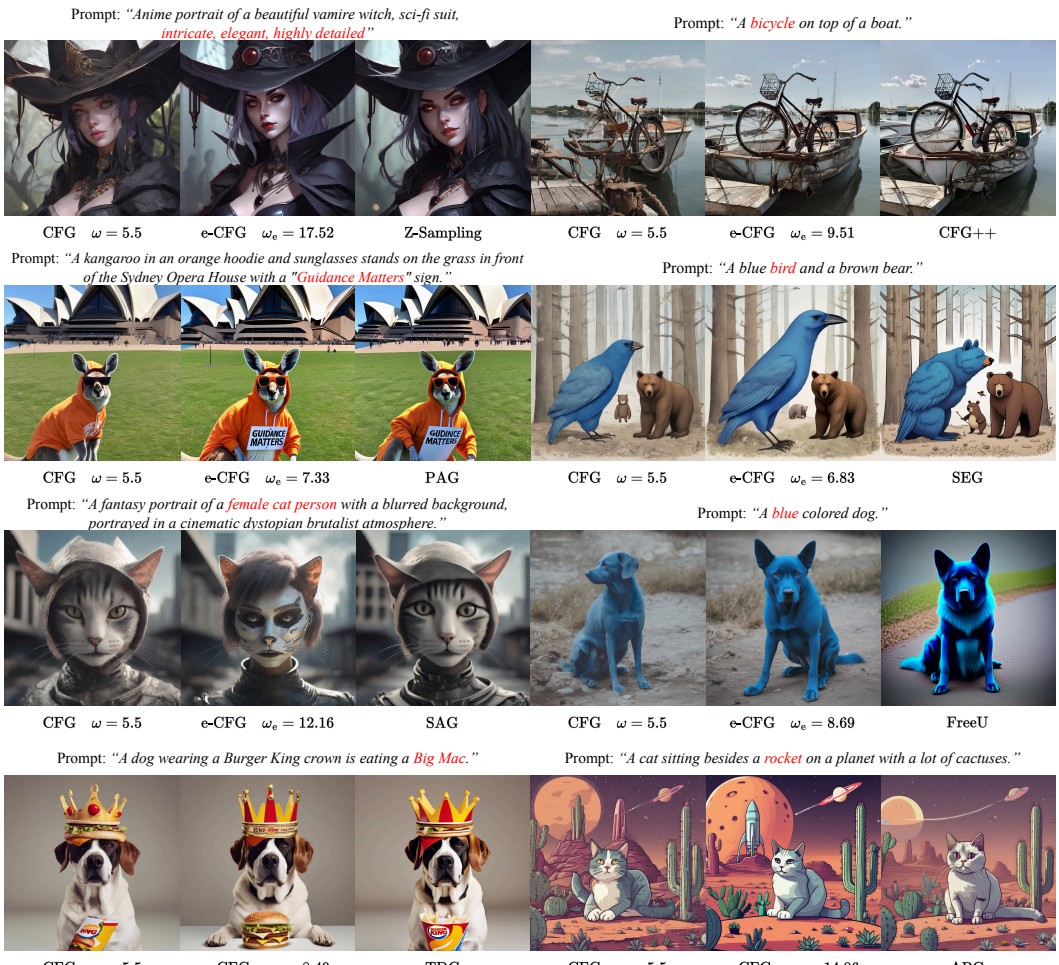

Figure 2: Visual comparison of different methods with their corresponding results under effective guidance scale $\omega^{\mathrm{e}}$. The e-CFG method, namely standard CFG with effective CFG scales calibrated by GA-Eval, can easily achieve performance improvement comparable to most recent diffusion guidance or sampling methods.

## 2 RELATED WORK

### 2.1 DIFFUSION MODEL

Diffusion model (Sohl-Dickstein et al., 2015) generate high-quality samples by gradually adding noise to the data and learning the inverse process. Denoising diffusion probabilistic models (DDPM) (Ho et al., 2020) applies this process to image generation. In the reverse process, a pure Gaussian noise was gradually transformed to original data:

$$\mathbf{x}_{t-1} = \frac{1}{\sqrt{\alpha_t}} \left( \mathbf{x}_t - \frac{\beta_t}{\sqrt{1 - \bar{\alpha}_t}} \epsilon_\theta \left( \mathbf{x}_t, t \right) \right) + \sigma_t \mathbf{z}, \tag{1}$$

where $\mathbf{z} \sim \mathcal{N}(0, \mathbf{I})$, $\beta_t = 1 - \alpha_t$, and $\epsilon_\theta$ is the noise prediction network. For simplicity, we will ignore the argument $t$ in $\epsilon_\theta(\mathbf{x}_t, t)$ in the following paragraphs.

To improve sampling efficiency, denoising diffusion implicit models (DDIM) Song et al. (2021a) use a non-Markovian sampling process that reduces the number of inference steps while maintaining generation quality. The reverse process can be written as:

$$\mathbf{x}_{t-1} = \sqrt{\alpha_{t-1}} \left( \frac{\mathbf{x}_t - \sigma_t \epsilon_\theta \left( \mathbf{x}_t \right)}{\sqrt{\alpha_t}} \right) + \sigma_{t-1} \epsilon_\theta \left( \mathbf{x}_t \right), \tag{2}$$

where $\sigma_t = \sqrt{1 - \alpha_t}$.

## 2.2 DIFFUSION GUIDANCE AND SAMPLING

Classifier-free guidance (CFG) (Ho & Salimans, 2021) interpolates conditional output and unconditional (or negative conditional) output of diffusion model, align the generated image to the corresponding condition. Given text prompt $\mathbf{c}$, CFG use a same network $\epsilon_\theta(\cdot)$ to predict the unconditional noise $\epsilon_t^{\text{uncond}} = \epsilon_\theta(\mathbf{x}_t, \varnothing)$. and the conditional noise $\epsilon_t^{\text{cond}} = \epsilon_\theta(\mathbf{x}_t, \mathbf{c})$. The updated noise in each denoising step was the interpolation between them under the guidance scale $\omega$:

$$\tilde{\epsilon}_t = \epsilon_t^{\text{uncond}} + \omega \left( \epsilon_t^{\text{cond}} - \epsilon_t^{\text{uncond}} \right). \tag{3}$$

Self-Attention Guidance (SAG) (Hong et al., 2023), Perturbed-Attention Guidance (PAG) (Ahn et al., 2024), and Smoothed Energy Guidance (SEG) (Hong, 2024) uses blurred attention-map or perturbed attention-map to create weak condidion term in CFG. CFG++ (Chung et al., 2025) aiming at tackling the off-manifold challenges inherent in traditional CFG. They reformulate text-guidance as an inverse problem with a text-conditioned score matching loss. Z-Sampling and W2SD (Bai et al., 2025a;b) use the guidance gap between denoising and inversion to inject semantic information during sampling process. FreeU (Si et al., 2024) amplified specific features in U-Net (Ronneberger et al., 2015) to improve performance for free. Although these methods have satisfying performance on current benchmarks, it can also achieved by enlarging the guidance scale. Kynkäänniemi et al. (2024) demonstrated that using CFG only when the noise intensity is within a certain range yields better results. Karras et al. (2024) used a degraded version of diffusion model to guide a complete one. But it is difficult to find a degraded model of a pretrained model. Adaptive Project Guidance (APG) (Sadat et al., 2025) project the latent parallel to the latent updated by conditional noise, which successfully eliminated over-saturation under large guidance scale. Nevertheless, it faces discrimination from many metrics, as they prefer images of high-saturation.

## 2.3 EVALUATION FOR TEXT-TO-IMAGE GENERATION

Traditional metrics for evaluating image quality include Inception Score (IS) (Salimans et al., 2016) and Fréchet Inception Distance (FID) (Heusel et al., 2017), but they cannot assess text-image alignment. Meanwhile, Jayasumana et al. (2024) have pointed out that FID and IS struggles to represent the rich content in modern image-to-text generation. The simplest way to evaluate images generated from a prompt is to use CLIP (Radford et al., 2021) to calculate text-image similarity, while it cannot reflect human preference. To address this problem, reward models are introduced to serve as metrics for human preference (Liu et al., 2024). HPS v2 (Wu et al., 2023), ImageReward (Xu et al., 2024), PickScore (Kirstain et al., 2023), MPS (Zhang et al., 2024), and Social Reward (Isajanyan et al., 2024) are evaluation metrics fine-tuned on human-preferenced data with reinforcement learning. These metrics are better aligned with human preferences, and have been widely used in benchmarking T2I models. Nevertheless, people naturally prefer colorful images, evaluation metrics based on human preferences tend to give higher scores to pictures with bright colors. Therefore, these metrics are easily blinded by images generated with a large guidance scale, as they also hold the same property of high-saturation. To the best of our knowledge, such evaluation pitfall have never been discussed before.

## 3 GUIDANCE MATTERS TO EVALUATION

In this section, we first demonstrate that the performance improvements brought by many guidance methods are largly exaggerated and similar to large CFG, and then propose the GA-Eval framework for more fair and reasonable evaluation.

**Bias of Large Guidance Scale.** We will present prevalent existence of the evaluation bias in this subsection. As some literatures (Bai et al., 2025a; Sadat et al., 2025) mentioned, diffusion models tend to generate over-saturated images under a large guidance scale, which will seriously affects the aesthetic quality of the image. Nevertheless, a larger guidance scale would amplify the predicted noise $\tilde{\epsilon}_t$ to the direction of conditional noise $\epsilon_t^{\text{cond}}$ in every timestep, thus making the generated image more aligned to the prompt.

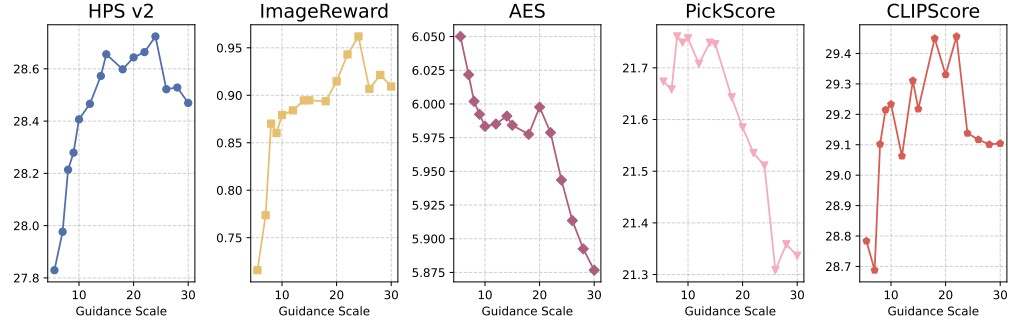

Figure 3: Different evaluation metrics under different guidance scales. Model: Stable Diffusion-XL. Dataset: Pick-a-Pic. Except AES and PickScore, other metrics would give higher ratings on images generated with a larger guidance scale within $\omega \in [5.5, 20]$.

However, this phenomenon could bias many evaluation metrics based on CLIP (Radford et al., 2021), as the CLIP model assumes the image is well-aligned to the prompt in this case. Moreover, for metrics fine-tuned on human-preferred data, the overlap between the training data and over-saturated results making these metrics easily biased by large guidance scale. We give case studies and quantitative results at Fig 1, Fig 2 and Fig 3. Most of T2I evaluation metrics would prefer images generated with a large guidance scale. Such bias may lead to results that do not reflect the real performance of other guidance methods, since we can simply increasing the guidance scale to acquire comparable or better performance on these metrics.

**Effective Guidance Scale.** Because better performance on can be achieved by increasing the guidance scale in CFG, we need a method to verify that whether a method's performance can be achieved by large guidance scale. Since this phenomenon was closely related to guidance scale, we can decouple the CFG component from other methods to calculate the effective guidance scale $\omega^{\mathrm{e}}$, thus verifying whether they exploit large guidance scale to achieve better performance.

For naive CFG in Eq 3, the updated noise in every timestep $\tilde{\epsilon}_t$ is the linear combination of the conditional noise $\epsilon_t^{\mathrm{cond}}$ and the unconditional noise $\epsilon_t^{\mathrm{uncond}}$. The guidance scale $\omega$ denoted as:

$$\omega = \frac{\tilde{\epsilon}_t - \epsilon_t^{\mathrm{uncond}}}{\epsilon_t^{\mathrm{cond}} - \epsilon_t^{\mathrm{uncond}}} = \frac{\tilde{\epsilon}_t - \epsilon_t^{\mathrm{uncond}}}{\Delta\epsilon}. \tag{4}$$

For other guidance methods, their noise updated in every timestep $\tilde{\epsilon}_t^*$ can be decomposed to $\epsilon_t^{\mathrm{uncond}}$ and two other components about $\Delta\epsilon$:

$$\tilde{\epsilon}_t^* = \epsilon_t^{\mathrm{uncond}} + \epsilon_t^{\|} + \epsilon_t^{\perp}, \tag{5}$$

where $\epsilon_t^{\perp}$ is the orthogonal component of $\tilde{\epsilon}_t^* - \epsilon_t^{\mathrm{uncond}}$ in the direction of $\Delta\epsilon$, and $\epsilon_t^{\|}$ is the parallel component of $\tilde{\epsilon}_t^* - \epsilon_t^{\mathrm{uncond}}$ in the direction of $\Delta\epsilon$. Since any vector can be decomposed into vectors orthogonal to each other.

According to the vector projection method, the orthogonal component $\epsilon_t^{\perp}$ can be calculated by:

$$\begin{aligned} \epsilon_t^{\perp} &= \left(\tilde{\epsilon}_t^* - \epsilon_t^{\mathrm{uncond}}\right) - \mathrm{Proj}_{\Delta\epsilon}\left(\tilde{\epsilon}_t^* - \epsilon_t^{\mathrm{uncond}}\right) \\ &= \left(\tilde{\epsilon}_t^* - \epsilon_t^{\mathrm{uncond}}\right) - \frac{\langle\tilde{\epsilon}_t^* - \epsilon_t^{\mathrm{uncond}}, \Delta\epsilon\rangle}{\|\Delta\epsilon\|^2}\Delta\epsilon. \end{aligned} \tag{6}$$

Therefore, the parallel component $\epsilon_t^{\|}$ is given by:

$$\epsilon_t^{\|} = \frac{\langle\tilde{\epsilon}_t^* - \epsilon_t^{\mathrm{uncond}}, \Delta\epsilon\rangle}{\langle\Delta\epsilon, \Delta\epsilon\rangle}\Delta\epsilon. \tag{7}$$

We define the effective guidance scale at timestep $t$ is the ratio of the amplitude of the parallel component $\epsilon_t^{\|}$ a to the amplitude of the guiding direction $\Delta\epsilon$.

$$\tilde{\epsilon}_t^* = \epsilon_t^{\mathrm{uncond}} + \omega_t^{\mathrm{e}}\Delta\epsilon + \epsilon_t^{\perp}, \tag{8}$$

Thus, the effective guidance scale can be acquired by:

$$\omega_t^{\mathrm{e}} = \frac{\|\epsilon_t^{\|}\|}{\|\Delta\epsilon\|}, \tag{9}$$

where $\|\cdot\|$ represents the $\ell_2$-norm.

However, some guidance methods (Bai et al., 2025a; Chung et al., 2025) modified the sampling process during the update of the latent $\mathbf{x}_t$, rather than explicitly modifying the predicted noise $\tilde{\epsilon}_t$. But these two approaches are interchangeable. For instance, in the case of DDIM from Eq 2, we can simply reformulate the process as follows:

$$\tilde{\epsilon}_t^* = \frac{\sqrt{\alpha_t}\mathbf{x}_{t-1} - \sqrt{\alpha_{t-1}}\mathbf{x}_t}{\sqrt{\alpha_t\beta_{t-1}} - \sqrt{\alpha_{t-1}\beta_t}}. \tag{10}$$

Therefore, we can acquire $\tilde{\epsilon}_t^*$ from the original latent $\mathbf{x}_t$ and the sampled latent $\mathbf{x}_{t-1}$. Note that the specific formulation of Eq 10 depends on the scheduler used by corresponding guidance method. Additionally, since $\omega_t^{\mathrm{e}}$ may vary across different timesteps $t$, we use a fixed effective guidance scale for each timestep by averaging along the sampling path: $\omega^{\mathrm{e}} = \frac{1}{T}\sum_t \omega_t^{\mathrm{e}}$.

**Guidance-aware Evaluation.** In this part, we introduce the GA-Eval framework that empirically evaluate diffusion guidance methods with effective guidance scales.

Although such bias could leads to an evaluation pitfall, we can use the bias itself to give a fair comparison with effective guidance scale. To determine whether a method's performance is exaggerated, we can measure how much the winning rate degrades after applying our settings to original CFG.

Given prompt set $\mathcal{C} = \{\mathbf{c}_i\}_{i=1}^N$, the images generated from each guidance method, CFG, and CFG with effective guidance scale denoted as $\left\{\left(\mathbf{X}_i^*, \mathbf{X}_i^{\mathrm{CFG}}, \mathbf{X}_i^{\mathrm{e\text{-}CFG}}\right)\right\}_{i=1}^N$, where they were generated with corresponding prompt $\mathbf{c}_i \in \mathcal{C}$. Denote binary comparison functions as follows:

$$
\begin{aligned}
\mathbb{I}_i^{\mathrm{CFG}} &= \begin{cases} 1 & \mathcal{M}(\mathbf{X}_i^*, \mathbf{c}_i) \succ \mathcal{M}(\mathbf{X}_i^{\mathrm{CFG}}, \mathbf{c}_i), \\ 0 & \text{otherwise}, \end{cases} \\
\mathbb{I}_i^{\mathrm{e\text{-}CFG}} &= \begin{cases} 1 & \mathcal{M}(\mathbf{X}_i^*, \mathbf{c}_i) \succ \mathcal{M}(\mathbf{X}_i^{\mathrm{e\text{-}CFG}}, \mathbf{c}_i), \\ 0 & \text{otherwise}, \end{cases}
\end{aligned}
\tag{11}
$$

where $\mathcal{M}$ is an evaluation metric, $\succ$ indicates the rating is better on the left side. The corresponding winning rate is given by:

$$\eta^{\mathrm{CFG}} = \frac{1}{N}\sum_{i=1}^N \mathbb{I}_i^{\mathrm{CFG}}, \quad \eta^{\mathrm{e\text{-}CFG}} = \frac{1}{N}\sum_{i=1}^N \mathbb{I}_i^{\mathrm{e\text{-}CFG}}. \tag{12}$$

By denoting the degradation of the winning rate, $\Delta\eta = \eta^{\mathrm{CFG}} - \eta^{\mathrm{e\text{-}CFG}}$, we can quantify whether a method just exploits a large guidance scale to achieve high performance.

## 4 TRANSCENDENT DIFFUSION GUIDANCE

In this section, we show that, motivated by the overlooked evaluation pitfall, we can easily design an effective TDG method that significantly increases the popular human preference metrics in the conventional evaluation framework but exposures the pitfall in the proposed GA-Eval framework.

Numerous recent works have demonstrated that use weak conditions or models can improve the generation performance of diffusion models, without any additional training or input. Like under-trained model (Karras et al., 2024), perturbed attention map Ahn et al. (2024); Hong et al. (2023), skipped layers (Hyung et al., 2025; Chen et al., 2025), etc.

Motivated by these recent works, we propose TDG to imitate the creation of weak condition. Specifically, TDG randomly replace tokens in text prompt $\mathbf{c}$ with empty token $\varnothing$, thus creating a weakened text prompt $\mathbf{c}^*$. Then use the noise prediction network $\epsilon_\theta(\cdot)$ to acquire the weak conditional score $\epsilon_{\mathrm{weak}}$, which combined with $\epsilon_{\mathrm{cond}}$ and $\epsilon_{\mathrm{uncond}}$ in CFG to the diffusion sampling process. We illustrate TDG in Fig 4 and present the pseudo-code in Algorithm 1. Due to page limits, we formally introduce TDG with more details and results in Appendix C.

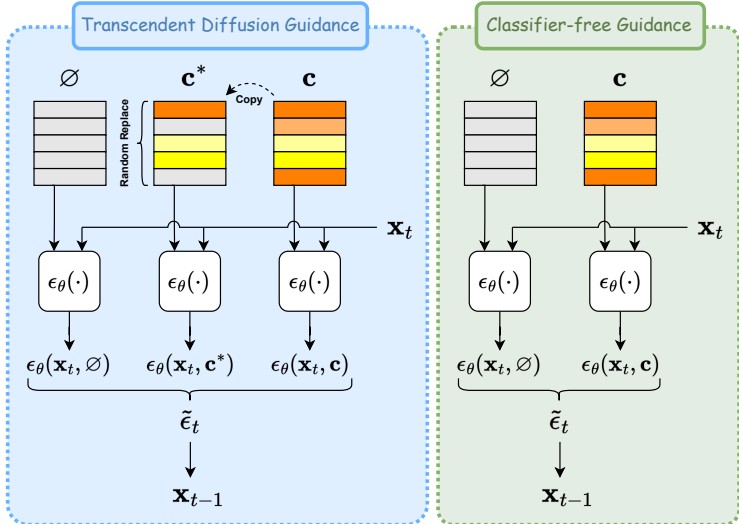

Figure 4: Comparison of classifier-free guidance Ho & Salimans (2021) and transcendent diffusion guidance (TDG).

## 5 EMPIRICAL ANALYSIS AND DISCUSSION

In this section, we empirically study how recent diffusion guidance methods perform in conventional text-to-image evaluation framework and our GA-Eval framework.

### 5.1 EXPERIMENTAL SETTINGS

**Models:** We choose Stable Diffusion-XL (Podell et al., 2024), Stable Diffusion 2.1 (Rombach et al., 2022), Stable Diffusion 3.5 (Esser et al., 2024), and DiT-XL/2 (Peebles & Xie, 2023) for image generation. We use $\omega = 5.5$, $T = 50$ for SD-XL and SD-2.1, $\omega = 4.5$, $T = 32$ for SD-3.5, and $\omega = 4$, $T = 50$ for DiT-XL/2.

**Datasets:** We use the first 100 prompts from Pick-a-Pic dataset (Kirstain et al., 2023), 200 prompts from DrawBench dataset (Saharia et al., 2022), Human Preferenced Dataset (HPD) (Wu et al., 2023) with 3200 prompts, and GenEval dataset (Ghosh et al., 2024) with 553 prompts. We also benchmarked some methods on COCO-30K (Lin et al., 2014). And ImageNet-50K (Deng et al., 2009) for the evaluation class-to-image generation.

**Evaluation Metrics:** We use HPS v2 (Wu et al., 2023), Aesthetics-Predictor (AES) (Schuhmann et al., 2022), PickScore (Kirstain et al., 2023), ImageReward (Xu et al., 2024), and CLIPScore (Radford et al., 2021) and Fréchet Inception Distance (FID) (Heusel et al., 2017), Inception Score (IS) (Barratt & Sharma, 2018; Salimans et al., 2016) for COCO-30K or ImageNet-50K.

**Baselines:** We select following guidance methods for comparison: Z-Sampling (Bai et al., 2025a), CFG++ (Chung et al., 2025), Perturbed-Attention Guidance (PAG) (Ahn et al., 2024), Self-Attention Guidance (SAG) (Hong et al., 2023), Smoothed-Energy Guidance (SEG) (Hong, 2024), FreeU (Si et al., 2024), Adaptive Project Guidance (APG) (Sadat et al., 2025), and Transcendent Diffusion Guidance (TDG) proposed in Sec 3. The detailed hyper-parameters of each method can be found at Sec B.

### 5.2 MAIN RESULTS

According to GA-Eval, almost all methods exaggerated their real performance which cannot be reflected in conventional evaluation paradigm. We present the results of eight diffusion guidance methods for SD-XL in Table 1. We also give an illustration of winning rate and its degradation on HPD dataset in Fig 5. Except APG that even cannot compete with standard CFG, most methods

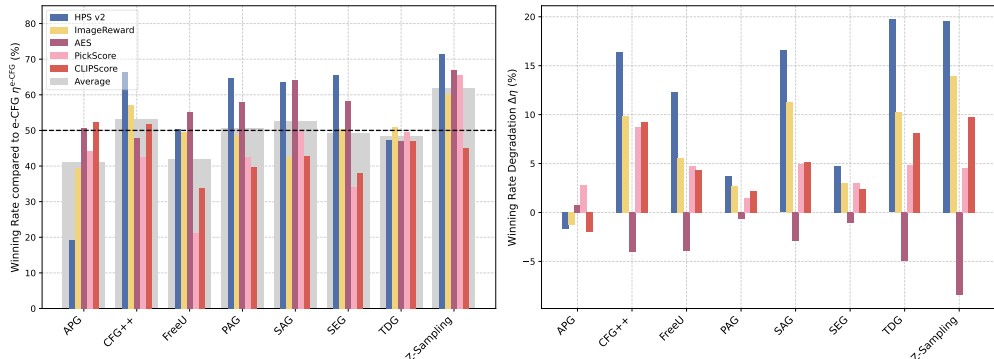

Figure 5: The winning rate $\eta^{\text{e-CFG}}$ compared to effective CFG and their degradation $\Delta\eta$ of different methods on HPD dataset. Left: $\eta^{\text{e-CFG}}$. Right: $\Delta\eta$. Among all methods, applying $\omega^{\text{e}}$ has minor influence to APG. Meanwhile, AES demonstrated negative $\Delta\eta$ on many methods, which means AES would give worse ratings to images generated with a large guidance scale.

Table 1: The winning rates $\eta^{\text{CFG}}, \eta^{\text{e-CFG}}$ and its degradation $\Delta\eta$ of different methods on various datasets. Model: Stable Diffusion-XL ($\omega = 5.5$). We sort these methods based on average $\eta^{\text{e-CFG}}$, labeling those greater than 55% in blue and the others in red. Most of methods have $\omega^{\text{e}} > \omega$, and suffering from severe winning rate degradation. Only recent Z-Sampling can still maintain high winning rates $\eta^{\text{e-CFG}}$ after setting CFG scale as $\omega^{\text{e}}$, while some methods even cannot compete with effective CFG.

| Method | HPS v2 | ImageReward | AES | PickScore | CLIPScore | Average $\eta$ | $\omega^{\text{e}}$ |
|---|---|---|---|---|---|---|---|
| **Pick-a-Pic** | | | $(\eta^{\text{CFG}}/\eta^{\text{e-CFG}}/\Delta\eta)$ | | | $(\eta^{\text{CFG}}/\eta^{\text{e-CFG}})$ | |
| Z-Sampling | 90% / 74% / 16% | 75% / 63% / 12% | 65% / 76% /-11% | 72% / 73% / -1% | 64% / 59% / 5% | 73% / 69% | 13.51 |
| CFG++ | 82% / 68% / 14% | 70% / 58% / 12% | 39% / 53% /-14% | 56% / 57% / -1% | 60% / 54% / 6% | 61% / 58% | 8.91 |
| SAG | 75% / 59% / 16% | 45% / 34% / 11% | 64% / 66% / -2% | 62% / 56% / 6% | 53% / 44% / 9% | 60% / 52% | 8.14 |
| TDG | 62% / 48% / 14% | 61% / 46% / 15% | 49% / 42% / 7% | 51% / 51% / 0% | 63% / 53% / 10% | 57% / 48% | 8.27 |
| SEG | 60% / 56% / 4% | 59% / 49% / 10% | 57% / 56% / 1% | 35% / 33% / 2% | 47% / 42% / 5% | 52% / 47% | 6.10 |
| PAG | 61% / 48% / 13% | 51% / 40% / 11% | 55% / 50% / 5% | 45% / 42% / 3% | 49% / 44% / 5% | 52% / 45% | 5.98 |
| FreeU | 52% / 42% / 10% | 56% / 44% / 12% | 53% / 55% / -2% | 27% / 25% / 2% | 42% / 33% / 9% | 46% / 40% | 7.47 |
| APG | 20% / 20% / 0% | 38% / 29% / 9% | 56% / 51% / 5% | 45% / 45% / 0% | 47% / 46% / 1% | 41% / 38% | 15.05 |
| **DrawBench** | | | $(\eta^{\text{CFG}}/\eta^{\text{e-CFG}}/\Delta\eta)$ | | | $(\eta^{\text{CFG}}/\eta^{\text{e-CFG}})$ | |
| Z-Sampling | 90% / 68% / 22% | 76% / 59% / 17% | 69% / 64% / 4% | 72% / 60% / 12% | 62% / 50% / 12% | 74% / 60% | 11.94 |
| PAG | 61% / 60% / 1% | 54% / 50% / 5% | 60% / 64% / -4% | 49% / 44% / 4% | 42% / 46% / -4% | 53% / 53% | 6.04 |
| TDG | 67% / 50% / 17% | 64% / 56% / 8% | 48% / 50% / -2% | 56% / 51% / 5% | 54% / 53% / 1% | 58% / 52% | 8.24 |
| SAG | 73% / 60% / 12% | 58% / 43% / 15% | 64% / 66% / -2% | 53% / 52% / 2% | 46% / 40% / 6% | 59% / 52% | 7.42 |
| CFG++ | 77% / 60% / 18% | 64% / 55% / 9% | 56% / 53% / 3% | 52% / 42% / 10% | 55% / 46% / 9% | 61% / 51% | 8.89 |
| SEG | 64% / 60% / 4% | 50% / 50% / 1% | 68% / 66% / 2% | 40% / 38% / 2% | 38% / 40% / -2% | 52% / 51% | 6.13 |
| FreeU | 48% / 37% / 11% | 54% / 46% / 8% | 64% / 62% / 2% | 25% / 22% / 2% | 46% / 45% / 1% | 47% / 43% | 7.44 |
| APG | 22% / 21% / 2% | 36% / 36% / 1% | 46% / 45% / 1% | 46% / 44% / 1% | 48% / 48% / -1% | 40% / 39% | 13.56 |
| **HPD** | | | $(\eta^{\text{CFG}}/\eta^{\text{e-CFG}}/\Delta\eta)$ | | | $(\eta^{\text{CFG}}/\eta^{\text{e-CFG}})$ | |
| Z-Sampling | 91% / 72% / 20% | 74% / 60% / 14% | 59% / 67% / -8% | 70% / 65% / 5% | 55% / 45% / 10% | 70% / 62% | 12.41 |
| CFG++ | 83% / 66% / 16% | 67% / 57% / 10% | 44% / 48% / -4% | 51% / 43% / 9% | 61% / 52% / 9% | 61% / 53% | 9.08 |
| SAG | 80% / 64% / 17% | 54% / 42% / 11% | 61% / 64% / -3% | 55% / 50% / 5% | 48% / 43% / 5% | 60% / 53% | 7.36 |
| PAG | 68% / 65% / 4% | 51% / 49% / 3% | 57% / 58% / -1% | 44% / 43% / 1% | 42% / 40% / 2% | 53% / 51% | 5.96 |
| SEG | 70% / 66% / 5% | 54% / 50% / 3% | 57% / 58% / -1% | 37% / 34% / 3% | 40% / 38% / 2% | 52% / 49% | 6.07 |
| TDG | 67% / 47% / 20% | 61% / 51% / 10% | 42% / 47% / -5% | 50% / 50% / 5% | 55% / 47% / 8% | 56% / 48% | 8.31 |
| FreeU | 63% / 50% / 12% | 55% / 50% / 6% | 51% / 55% / -4% | 26% / 21% / 5% | 38% / 34% / 4% | 47% / 42% | 7.38 |
| APG | 18% / 19% / -2% | 38% / 39% / -1% | 51% / 51% / 1% | 47% / 44% / 3% | 50% / 52% / -2% | 41% / 41% | 13.29 |

suffer from significant winning rate degradations over effective CFG in GA-Eval. The HPS v2 degradation in CFG++, SAG, TDG, and Z-Sampling are even greater than 15%. Most of baselines, except Z-Sampling, have the averaged $\eta^{\text{e-CFG}}$ near or worse than 50%. In the experiment of SD-2.1, Table 4 in Sec 4, our GA-Eval also reveals the prevalently overlooked evaluation pitfall. We also use GenEval to evaluate the fine-grained property and semantic alignment. In Table 2, most methods are worse than effective CFG in terms of the average GenEval scores.

Despite suffering winning rate degradations, some methods still maintained a considerable winning rate after applying $\omega^{\text{e}}$. In Table 1, the winning rate of HPS v2 of Z-Sampling is approximately

Table 2: The quantitative results of different methods on GenEval. Model: Stable Diffusion-XL ($\omega = 5.5$). We labeled methods surpassed effective CFG with blue, and others with red. Increasing the CFG scale can also significantly increase the GenEval scores.

| Method | Single object (↑) | Two object (↑) | Counting (↑) | Colors (↑) | Position (↑) | Color attribution (↑) | Overall (↑) | $\omega^e$ |
|---|---|---|---|---|---|---|---|---|
| CFG | 98.91% | 75.63% | 37.66% | 85.11% | 10.87% | 26.88% | 55.84% | 5.5 |
| TDG | 100.00% | 76.77% | 53.75% | 87.23% | 11.00% | 21.00% | 58.29% | 8.28 |
| e-CFG | 100.00% | 73.74% | 47.50% | 86.17% | 9.00% | 27.00% | 57.23% | |
| Z-Sampling | 100.00% | 75.76% | 46.25% | 84.04% | 17.00% | 20.00% | 57.18% | 16.05 |
| e-CFG | 98.75% | 77.78% | 36.25% | 85.11% | 16.00% | 25.00% | 56.48% | |
| CFG++ | 100.00% | 81.82% | 46.25% | 85.11% | 16.00% | 20.00% | 58.20% | 8.81 |
| e-CFG | 100.00% | 76.77% | 48.75% | 87.23% | 12.00% | 24.00% | 58.13% | |
| FreeU | 98.75% | 74.75% | 41.25% | 76.60% | 9.00% | 25.00% | 54.22% | 7.62 |
| e-CFG | 100.00% | 76.77% | 48.75% | 88.30% | 12.00% | 25.00% | 58.47% | |
| PAG | 93.75% | 69.70% | 37.50% | 82.98% | 9.00% | 26.00% | 53.15% | 6.13 |
| e-CFG | 100.00% | 76.77% | 42.50% | 85.11% | 9.00% | 28.00% | 56.90% | |
| SAG | 97.50% | 75.76% | 46.25% | 86.17% | 9.00% | 25.00% | 56.61% | 7.36 |
| e-CFG | 100.00% | 78.79% | 43.75% | 86.17% | 10.00% | 26.00% | 57.45% | |
| SEG | 97.50% | 71.72% | 41.25% | 81.91% | 9.00% | 18.00% | 53.23% | 6.19 |
| e-CFG | 98.75% | 77.78% | 41.25% | 84.04% | 9.00% | 26.00% | 56.14% | |
| APG | 98.75% | 70.71% | 40.00% | 87.23% | 12.00% | 27.00% | 55.95% | 14.12 |
| e-CFG | 100.00% | 78.79% | 46.25% | 89.36% | 12.00% | 31.00% | 59.57% | |

70%, where the averaged winning rate is greater than 60% across different datasets. CFG++ also has certain but lower superiority.

Meanwhile, many metrics obviously biased by large guidance scale. Given $\omega^e > \omega$, most metrics have $\Delta\eta > 0$, which means they raised ratings to images from effective CFG with large guidance scale. Among these metrics, only AES have reversed results. GenEval also demonstrated the evaluation pitfall brought by large guidance scales.

Besides, we conduct experiments on COCO-30K and ImageNet-50K with Z-Sampling and CFG++. Due to page limit, the results are presented in Table 8 and Table 9 in Sec D. After applying $\omega^e$, the performance of e-CFG are getting closer to the corresponding method. However, they are still worse than the original CFG in FID and IS.

## 5.3 DISCUSSION AND ANALYSIS

**Evaluation Metrics.** The results in various metrics in Table 1 align with our observations in Fig 3. HPSv2, ImageReward, and PickScore are largely affected by large guidance scales. This might be because they were trained on human-preferenced data with high saturated images well aligned to the prompt, which are very similar to images generated with a large guidance scale. For CLIPScore, its semantic alignment may also contribute to such bias, as large guidance scales naturally improve image-prompt alignment. In GenEval, large guidance scale also contributes to the increased semantic correctness of object-level properties. Although most metrics are biased, AES provides a fair evaluation, since it only evaluates images. However, this also makes it unable to evaluate prompt-following abilities in generated images. The AIGC community eagerly needs a human preference model robust to large CFG.

**Guidance Methods.** Since most evaluation metrics are biased, CFG with larger guidance scale of course bring severe winning rate degradations to these studied methods. In principle, many methods implicitly amplified their effective guidance scale. Z-Sampling uses a normal guidance scale in denoising, and a smaller guidance scale in inversion. The guidance gap between denoising and inversion indirectly amplified the guidance scale. For methods like PAG, SAG, SEG, and TDG, which created an additional weak condition term to the updated noise in each timestep, the difference of conditional noise and weak conditional noise is parallel to $\Delta\epsilon$ with a significant portion, which result a larger $\omega^e$. The formulation of CFG++ also increased the guidance scale compared to original CFG. Notably, APG holds a very low winning rate before and after apply $\omega^e$, with only AES maintained normal. It is because APG alleviate over-saturation in generated images, other biased metrics would give worse evaluation ratings. So these metrics cannot reflect the real performance of APG. APG

did not exploit large guidance scale, although it explicitly uses large CFG scales $\omega = 15$ for SD-XL and $\omega = 10$ for SD-21.

For methods maintained effectiveness compared to effective CFG, like Z-Sampling and CFG++. This indicates that the performance improvement of these methods is not solely achieved by increasing the guidance scale. They indeed have some effective components orthogonal to CFG. Meanwhile, their FID and IS also indicate that their distribution are shifted away from ground-truth compared to CFG, since the later was trained on ground-truth images with a fixed guidance scale.

**Diffusion Models.** Not limited to results of SD-XL in Table 1, we have conducted experiments on other models as well. The results of SD-2.1 and SD-3.5 are presented in Table 4, and Table 3 in Sec D. In SD-2.1, for CFG++ and Z-Sampling, their winning rate degradations are even larger than SD-XL (above 20%). For simple models like SD-2.1, it is quite easier to rectify the sampling trajectory to improve the generation quality, although they often collapse to a larger guidance scale. For SD-3.5, its generation capability is already very powerful, which limits the improvements due to guidance methods.

## 6 CONCLUSION

In this work, we report a critical but overlooked evaluation pitfall hidden in recent advances in diffusion sampling and its evaluation for text-to-image generation. These pitfalls severely damage development of this line of research. As human preference models exhibit strong bias to large guidance scales, we are the first to reveal that most recent diffusion guidance methods even cannot outperform standard CFG with large guidance scales. To more properly evaluate diffusion guidance methods, we propose a novel GA-Eval framework that disentangles the effects orthogonal and parallel to the CFG effect. Motivated by the findings, we also propose TDG to have improved evaluation scores in the conventional evaluation framework. Through comprehensive experiments in the GA-Eval framework, we expose that most claimed improvements, including TDG, essentially mirror the effects of simply increasing the CFG scale, with all methods showing inflated performance metrics under proper evaluation. We believe that this work serves as a crucial wake-up call for the community to rethink the evaluation paradigm for AIGC tasks and more properly recognize true innovation in diffusion models.

ACKNOWLEDGMENTS

This work was supported by the Science and Technology Bureau of Nansha District Under Key Field Science and Technology Plan Program No. 2024ZD002 and Dream Set Off - Kunpeng and Ascend Seed Program.

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

## A  LLM USAGE

In this paper, no Large Language Model (LLM) was employed as a research or writing tool. All research conception, data analysis, writing, and editing were independently accomplished by the author.

## B  EXPERIMENTAL SETTINGS

All our experiments were conducted on NVIDIA GeForce RTX 4090 GPUs with 24GB VRAM.

In SD-XL, we use guidance scale $\omega = 5.5$ and inference step $T = 50$. For Z-Sampling, we use inversion guidance scale $\omega_{\text{inv}} = 0$, optimization steps $\lambda = T$. For PAG, we use PAG scale $s = 3$, and apply PAG on layer "mid" of the backbone of SD-XL. For CFG++, we use guidance scale $\omega = 0.4$. For TDG, we use $g = 1.8$ and $\beta = 2.6$. For APG, we use guidance scale $\omega = 15$. For FreeU, we use $b_1 = 1.3$, $b_2 = 1.4$, $s_1 = 0.9$, and $s_2 = 0.2$. For SAG, we use SAG scale $s = 0.75$. For SEG, we use SEG scale $\gamma_{\text{seg}} = 3$, standard deviation of Gauusian blur $\sigma = 100$, and apply on layer "mid" of the backbone of SD-XL.

In SD-2.1, we use guidance scale $\omega = 5.5$ and inference step $T = 50$. For Z-Sampling, we use inversion guidance scale $\omega_{\text{inv}} = 0$, optimization steps $\lambda = T$. For PAG, we use PAG scale $s = 3$, and apply PAG on layer "mid" of the backbone of SD-XL. For CFG++, we use guidance scale $\omega = 0.4$. For FreeU, we use $b_1 = 1.4$, $b_2 = 1.6$, $s_1 = 0.9$, and $s_2 = 0.2$. For SAG, we use SAG scale $s = 0.75$.

In SD-3.5, we use 4-bit NormalFloat quantization to run on RTX-4090, with guidance scale $\omega = 4.5$ and inference step $T = 32$. For Z-Sampling, we use inversion guidance scale $\omega_{\text{inv}} = 0$, optimization steps $\lambda = T$. For PAG, we use PAG scale $s = 3$, and apply PAG on layer "13" of the backbone of SD-3.5. For TDG, we use $g = 1.6$ and $\beta = 2.4$.

In DiT-XL/2, we use guidance scale $\omega = 4$ and timestep $T = 50$. For Z-Sampling, we use inversion guidance scale $\omega_{\text{inv}} = 0$, optimization steps $\lambda = T$. For CFG++, we use guidance scale $\omega = 0.5$.

## C  TRANSCEDENT DIFFUSION GUIDANCE

In this section, we will present limitations of previous guidance techniques. After that, we will introduce the proposed method transcendent diffusion guidance (TDG), and illustrate how can we improve the sampling process of diffusion models by enlarging search space.

### C.1  MOTIVATION

Previous improvements to CFG have been limited to the guidance scale $\omega$ or use self-defined functions to substitute $\epsilon_{\text{uncond}}$ or $\epsilon_{\text{cond}}$. However, these methods are limited to the operation in Eq 3: interpolation between the two outputs of the diffusion model.

To solve this problem, we need to expand the search space of the sampling process. Obviously, due to the search space of CFG restricted on a line, we only need to have a point outside of this line to obtain the whole hyperplane as the search space. Besides, the point should be acquired through the text prompt **c** without introducing any additional inputs.

Driven by the analysis above, we propose transcendent diffusion guidance to improve the sampling process of diffusion models, which illustrated in Fig. 4. Specifically, TDG corrupts the text prompt **c** to create a weakened text prompt **c**$^*$. Then use the noise prediction network $\epsilon_\theta(\cdot)$ to acquire the weak conditional score $\epsilon_{\text{weak}}$. Combining with $\epsilon_{\text{cond}}$ and $\epsilon_{\text{uncond}}$ in CFG, we can expand the search space to the entire hyperplane. During the sampling process, larger search space will accelerate the convergence of latent towards the optimal solution, thereby improving the generation quality of the model.

### C.2  WEAK CONDITION

In this subsection, we will provide a detailed explanation on how to obtain a weak condition.

---

**Algorithm 1** The Sampling of Transcendent Diffusion Guidance

---

**Input:** text condition $\mathbf{c}$, guidance scale factor $g$, balance scale factor $\beta$, random replace ratio $\lambda$

$\mathbf{c}^* = \text{random\_replace}(\mathbf{c}, \lambda)$

$\mathbf{x}_T \sim \mathcal{N}(0, \mathbf{I})$

**for** $t$ in $T, T-1, \ldots, 1$ **do**

$\quad \epsilon_{\text{uncond}}^t = \epsilon_\theta(\mathbf{x}_t, \varnothing)$

$\quad \epsilon_{\text{cond}}^t = \epsilon_\theta(\mathbf{x}_t, \mathbf{c})$

$\quad \epsilon_{\text{weak}}^t = \epsilon_\theta(\mathbf{x}_t, \mathbf{c}^*)$

$\quad \tilde{\epsilon}_t = \frac{1}{2}(\epsilon_t^{\text{uncond}} + \epsilon_t^{\text{weak}}) + \frac{\omega \cdot g \cdot \beta}{\beta+1}(\epsilon_t^{\text{cond}} - \epsilon_t^{\text{uncond}}) + \frac{\omega \cdot g}{\beta+1}(\epsilon_t^{\text{cond}} - \epsilon_t^{\text{weak}}) \cdot \frac{\left\| \epsilon_t^{\text{cond}} - \epsilon_t^{\text{uncond}} \right\|}{\left\| \epsilon_t^{\text{cond}} - \epsilon_t^{\text{weak}} \right\|}$

$\quad \mathbf{x}_{t-1} \sim \mathcal{N}\left(\frac{1}{\sqrt{\alpha_t}}(\mathbf{x}_t - \frac{\beta_t}{\sqrt{1-\bar{\alpha}_t}}\tilde{\epsilon}_t), \Sigma_t\right)$

**end for**

**return** $\mathbf{x}_0$

---

As mentioned in the main text, many works demonstrated that introduce an additional weak model or weak condition could improve the generation performance. Meanwhile, they should avoid introducing any additional input and be computationally efficient. Therefore, we obtain the weak condition outside of CFG by modifying the text prompt. Specifically, consider a text prompt $\mathbf{c} = [c_1, c_2, \ldots, c_n]$, where $c_i$ is the $i$-th token of $\mathbf{c}$. We randomly choose some tokens with index $i \in \mathcal{I}$, and replace them with empty token $\varnothing$:

$$c_i^* = \begin{cases} \varnothing, & i \in \mathcal{I} \\ c_i, & i \notin \mathcal{I} \end{cases}. \tag{13}$$

By combining these new tokens together, we obtained the weakened prompt $\mathbf{c}^* = [c_1^*, \ldots, c_n^*]$. In practice, $|\mathcal{I}| = \frac{n}{2}$, which means we randomly replace half tokens in $\mathbf{c}$ with $\varnothing$. On the one hand, the modified prompt only requires the original prompt to be obtained without introducing additional input. On the other hand, to obtain $\mathbf{c}^*$, it only needs to be calculated at the beginning, which is computationally efficient compared to those methods need to manipulating attention maps at every denoising step.

Therefore, the weak conditional score is given by:

$$\epsilon_t^{\text{weak}} = \epsilon_\theta(\mathbf{x}_t, \mathbf{c}^*). \tag{14}$$

## C.3 ALGORITHM

After obtaining the weak condition, we can construct the hyperplane through an extra guidance term. Compared to CFG in Eq 3, TDG reformulated the diffusion score in each denoising step with following add-on:

$$\tilde{\epsilon}_t = \frac{1}{2}(\epsilon_t^{\text{uncond}} + \epsilon_t^{\text{weak}}) + \frac{\omega \cdot g \cdot \beta}{\beta+1}(\epsilon_t^{\text{cond}} - \epsilon_t^{\text{uncond}}) +$$
$$\frac{\omega \cdot g}{\beta+1}(\epsilon_t^{\text{cond}} - \epsilon_t^{\text{weak}}) \cdot \frac{\left\| \epsilon_t^{\text{cond}} - \epsilon_t^{\text{uncond}} \right\|}{\left\| \epsilon_t^{\text{cond}} - \epsilon_t^{\text{weak}} \right\|}, \tag{15}$$

where $g$ and $\beta$ are guidance scale factor and balance scale factor, respectively. For noise prediction network $\epsilon_\theta(\cdot)$, the linear combination of conditional score $\epsilon_t^{\text{cond}}$, the weak conditional score $\epsilon_t^{\text{weak}}$, and the unconditional score $\epsilon_t^{\text{uncond}}$ will create three non-collinear points, thus creating a hyperplane. While the original CFG only limits its search space the combination of $\epsilon_t^{\text{cond}}$ and $\epsilon_t^{\text{uncond}}$, which is a straight line. Therefore, the model can have a hyperplane as the search space, which transcend the restrictions of the original search space of CFG. We give a brief illustration in Fig 8. And the hyperparameter grid search result was illustrated in Fig 9.

The pseudo-code is provided in Alg 1. Fig 4 also briefly illustrates the differences between TDG and CFG. Qualitative comparison results are presented in Fig 6. The hyperparameters of TDG are determined through grid search, and the criteria is HPSv2 winning rate compared to original CFG. The results can be found in Fig 9. Meanwhile, we give an analysis on the number of inference step $T$ to demonstrate the superiority of TDG. In Fig 7, TDG with $T = 16$ achieved comparable performance on CFG with $T = 20$, and TDG always maintain better performance than CFG when

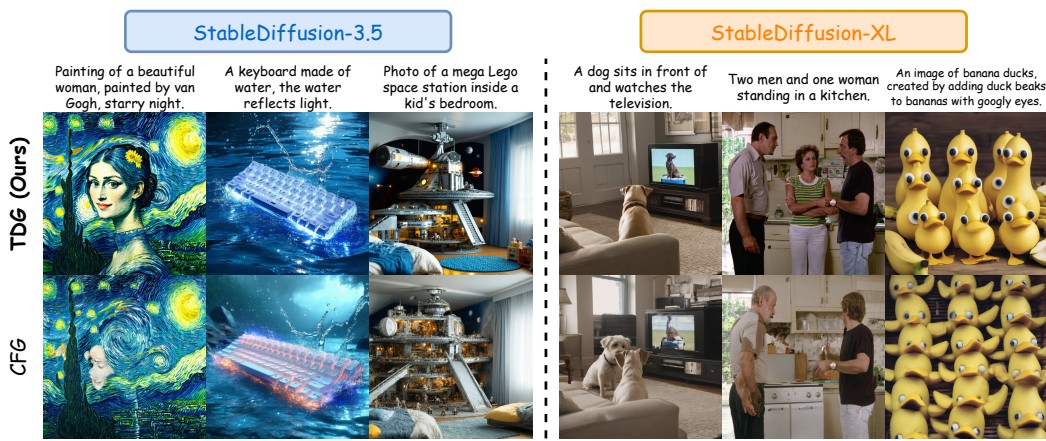

Figure 6: The qualitative results of TDG.

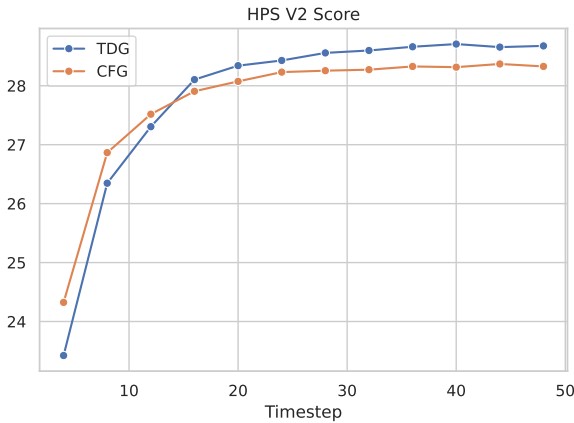

Figure 7: Comparison of HPS v2 Score of TDG and CFG at different timesteps. Model: SD-3.5.

$T \geq 16$. We also present ablation results of TDG in Fig 10. There is no significant performance degradation when mask ratio smaller than 0.6.

## D  SUPPLEMENTARY EMPIRICAL RESULTS

We give more cases of HPS v2 scores increasing with $\omega$ in Fig 11, the phenomenon can be verified across different prompts, and has further verified in Fig 3. For simplicity, we give the quantitative performance of SD-XL, SD-2.1, and SD-3.5 are given in Table 5, Table 6, and Table 7, respectively. Specifically, the winning rates and the degradation of different methods on SD-3.5 is given in Table 3. The results are consistent with our findings in the maintext. More cases of visual comparison between other methods and their corresponding effective CFG are presented below.

In addition, we test recent reward models and benchmark, like HPSv3 (Ma et al., 2025), UnifiedReward (Wang et al., 2025), and OneIG-Bench (Chang et al., 2025), to valid their robustness on the evaluation pitfall. And the results are presented in Table 10, 11, and 12. HPSv3 still hold this evaluation pitfall, with most methods suffer winning rate degradation. Meanwhile, due to OneIG-Bench require to use theirs prompts to evaluate, we only conduct experiments on SD-XL with different guidance scales. The Alignment score, Reasoning score, and Text score are increasing with guidance scale. But Diversity score decreases on the contrary, this also meet the discovery in (Karras et al., 2024).

To further demonstrate that human-preferred metrics favor saturated images, we give 2 toy experiments on images with different saturation: 1) Real images with different saturation; 2) Generated

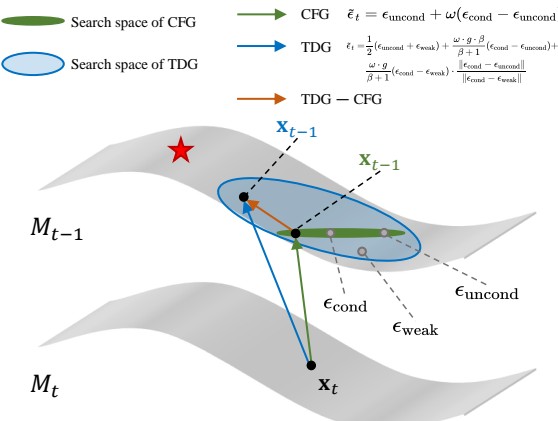

Figure 8: TDG vs. CFG in manifold space. Compared to CFG, TDG enlarges the search space to a hyperplane.

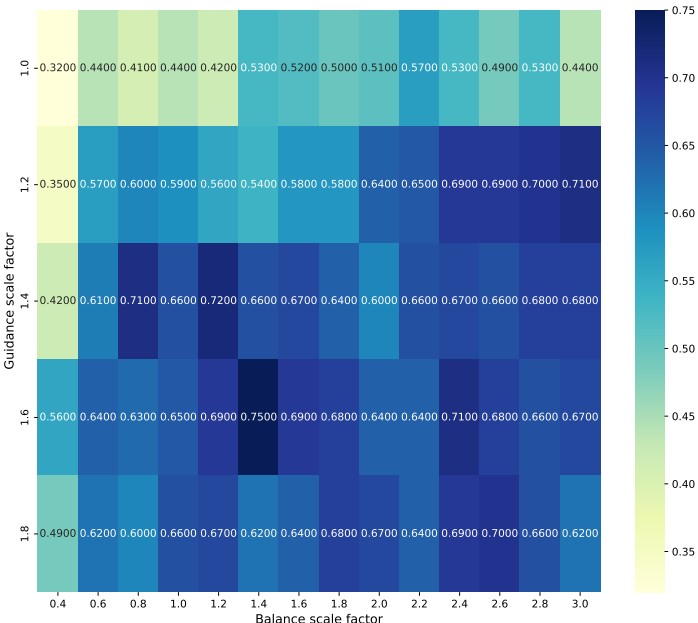

Figure 9: TDG hyperparameter grid search heatmap, with HPSv2 winning rate compared to CFG as the criteria. Model: SD-XL.

images with different guidance scale; We also use Spearman' test to valid the correlations between the image saturation and metric score, the alpha value was set to 0.05.

- 1) For real images, we randomly selected 20 Ground-Truth images from MS-COCO dataset, and changed their saturation with different levels. The scatter map are presented in Fig 12. Most prompts have metric score positively correlated with image saturation. In HPSv2, Significant positive correlations: 14, Positive correlation not significant: 1, No significant positive correlation: 5. In ImageReward, Significant positive correlations: 14, Positive correlation not significant: 2, No significant positive correlation: 4.

- 2) For generated images, we randomly select 20 prompts from Pick-a-Pic, DrawBench, and HPD. Then generated then with different guidance scales. The scatter map are presented in

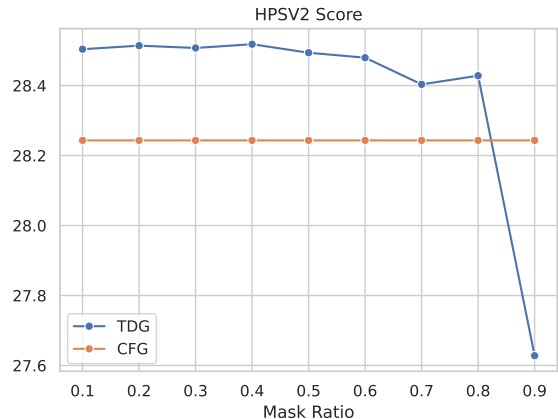

Figure 10: Different mask ratio of prompt tokens in TDG. Model: SD-XL.

Table 3: The winning rate and its degradation of different methods on various datasets. Model: Stable Diffusion-3.5 ($\omega = 4.5$).

| Method | HPS v2 | ImageReward | AES | PickScore | CLIPScore | Average $\eta$ | $\omega^e$ |
|---|---|---|---|---|---|---|---|
| **Pick-a-Pic** | | | $(\eta^{\text{CFG}}/\eta^{\text{e-CFG}}/\Delta\eta)$ | | | $(\eta^{\text{CFG}}/\eta^{\text{e-CFG}})$ | |
| PAG | 53% / 53% / 0% | 44% / 43% / 1% | 53% / 57% / -4% | 36% / 43% / -7% | 44% / 41% / 3% | 46% / 47% | 5.33 |
| TDG | 61% / 48% / 13% | 50% / 51% / -1% | 45% / 51% / -6% | 38% / 47% / -9% | 53% / 47% / 6% | 49% / 49% | 6.62 |
| Z-Sampling | 40% / 36% / 4% | 41% / 54% /-13% | 46% / 49% / -3% | 31% / 37% / -6% | 53% / 58% / -5% | 42% / 47% | 9.57 |
| **DrawBench** | | | $(\eta^{\text{CFG}}/\eta^{\text{e-CFG}}/\Delta\eta)$ | | | $(\eta^{\text{CFG}}/\eta^{\text{e-CFG}})$ | |
| PAG | 51% / 50% / 2% | 44% / 40% / 3% | 61% / 64% / -3% | 42% / 40% / 2% | 39% / 42% / -3% | 48% / 47% | 5.41 |
| TDG | 57% / 46% / 12% | 54% / 45% / 9% | 47% / 48% / -1% | 41% / 44% / -4% | 48% / 49% / -1% | 50% / 46% | 6.59 |
| Z-Sampling | 50% / 41% / 9% | 50% / 45% / 4% | 51% / 47% / 4% | 34% / 44% / -9% | 46% / 48% / -1% | 46% / 45% | 9.34 |

Fig 13. And the problem still remains: In HPSv2, Significant positive correlations: 15, Positive correlation not significant: 2, No significant positive correlation: 3. In ImageReward, Significant positive correlations: 12, Positive correlation not significant: 1, No significant positive correlation: 7.

We analyzed the $\omega_t^e$ distribution over timestep $t$. In Fig 14 (a), every method have their own $\omega_t^e$ distribution on $t$, but the $\omega_t^e$ averaged on $t$ is still greater than the baseline $\omega = 5.5$. In Fig 14 (b), the different scheduler have minor effect on the $\omega_t^e$.

Prompt: *"A beautiful blonde woman with freckles eating desserts with a dog at a dining table. The background of Eiffel Tower with neon fireworks lighting up the night sky."*

| HPS v2: 29.7943 | HPS v2: 30.5297 | HPS v2: 30.3956 | HPS v2: 30.7722 |

Prompt: *"A cat, chubby, very fine wispy and extremely long swirly wavy fur, under water, Kuniyoshi Utagawa, Hishida Shuns."*

| HPS v2: 28.6557 | HPS v2: 29.2244 | HPS v2: 29.6573 | HPS v2: 29.7427 |

Prompt: *"A brown colored giraffe."*

| HPS v2: 29.6685 | HPS v2: 29.9727 | HPS v2: 30.2884 | HPS v2: 30.3286 |

Figure 11: The HPS v2 scores of generated images under different CFG scales $\omega \in \{5.5, 10, 15, 20\}$, respectively. Model: Stable Diffusion-XL. HPSv2 exhibit a strong bias to large CFG scales in a wide range, even if generation quality starts to degrade due to too strong guidance.

Table 4: The winning rates and the degradation of different methods on various datasets. Model: Stable Diffusion-2.1 ($\omega = 5.5$).

| Method | HPS v2 | ImageReward | AES | PickScore | CLIPScore | Average $\eta$ | $\omega^{\mathrm{e}}$ |
|---|---|---|---|---|---|---|---|
| **Pick-a-Pic** | $(\eta^{\mathrm{CFG}}/\eta^{\mathrm{e\text{-}CFG}}/\Delta\eta)$ | | | | | $(\eta^{\mathrm{CFG}}/\eta^{\mathrm{e\text{-}CFG}})$ | |
| Z-Sampling | 90% / 60% / 30% | 80% / 62% / 18% | 70% / 63% / 7% | 84% / 73% / 11% | 71% / 46% / 25% | 79% / 61% | 11.80 |
| CFG++ | 82% / 62% / 20% | 73% / 58% / 15% | 58% / 55% / 3% | 65% / 49% / 16% | 57% / 55% / 2% | 67% / 56% | 9.09 |
| FreeU | 64% / 61% / 3% | 73% / 67% / 6% | 63% / 57% / 6% | 52% / 45% / 7% | 50% / 47% / 3% | 60% / 55% | 6.57 |
| PAG | 48% / 45% / 3% | 49% / 41% / 8% | 54% / 55% / -1% | 49% / 45% / 4% | 46% / 44% / 2% | 49% / 46% | 5.62 |
| APG | 11% / 31% /-20% | 21% / 43% /-22% | 49% / 53% / -4% | 26% / 34% / -8% | 45% / 58% /-13% | 30% / 44% | 11.19 |
| SAG | 42% / 28% / 14% | 52% / 37% / 15% | 47% / 46% / 1% | 30% / 23% / 7% | 56% / 39% / 17% | 45% / 35% | 6.53 |
| **DrawBench** | $(\eta^{\mathrm{CFG}}/\eta^{\mathrm{e\text{-}CFG}}/\Delta\eta)$ | | | | | $(\eta^{\mathrm{CFG}}/\eta^{\mathrm{e\text{-}CFG}})$ | |
| FreeU | 76% / 68% / 8% | 68% / 64% / 5% | 67% / 65% / 2% | 57% / 54% / 3% | 50% / 48% / 2% | 64% / 60% | 6.65 |
| Z-Sampling | 88% / 62% / 25% | 73% / 65% / 8% | 66% / 52% / 14% | 74% / 66% / 8% | 65% / 52% / 13% | 73% / 59% | 11.74 |
| CFG++ | 80% / 62% / 17% | 70% / 57% / 12% | 67% / 66% / 2% | 58% / 48% / 11% | 60% / 56% / 4% | 67% / 58% | 9.08 |
| SAG | 42% / 30% / 11% | 54% / 42% / 11% | 62% / 66% / -4% | 38% / 38% / 1% | 55% / 48% / 7% | 50% / 45% | 6.45 |
| PAG | 46% / 42% / 4% | 42% / 42% / 1% | 50% / 45% / 4% | 46% / 48% / -1% | 42% / 43% / -2% | 45% / 44% | 5.63 |
| APG | 16% / 29% /-13% | 30% / 42% /-12% | 38% / 48% /-10% | 28% / 40% /-12% | 38% / 44% / -6% | 30% / 41% | 10.48 |

Table 5: Quantitative results of different methods. Model: Stable Diffusion-XL ($\omega = 5.5$).

| Dataset | Method | HPS v2 (↑) | ImageReward (↑) | AES (↑) | PickScore (↑) | CLIPScore (↑) |
|---|---|---|---|---|---|---|
| **Pick-a-Pic** | CFG | 28.51 | 0.845 | 6.112 | 22.50 | 31.01 |
| | APG | 28.70 | 0.937 | 5.995 | 22.53 | 32.28 |
| | e-CFG | 29.30 | 1.032 | 5.992 | 22.61 | 32.23 |
| | CFG++ | 29.21 | 1.025 | 6.091 | 22.60 | 31.46 |
| | e-CFG | 29.05 | 0.990 | 6.103 | 22.64 | 31.39 |
| | FreeU | 28.88 | 0.929 | 6.156 | 22.11 | 30.01 |
| | e-CFG | 28.90 | 0.955 | 6.115 | 22.63 | 31.23 |
| | PAG | 28.92 | 0.863 | 6.174 | 22.47 | 30.54 |
| | e-CFG | 28.68 | 0.884 | 6.126 | 22.56 | 31.09 |
| | SAG | 28.87 | 0.884 | 6.087 | 22.55 | 30.97 |
| | e-CFG | 28.70 | 0.931 | 6.022 | 22.55 | 31.26 |
| | SEG | 28.96 | 0.875 | 6.178 | 22.35 | 30.47 |
| | e-CFG | 28.70 | 0.893 | 6.123 | 22.56 | 31.12 |
| | TDG | 28.91 | 0.972 | 6.092 | 22.61 | 31.19 |
| | e-CFG | 28.96 | 0.955 | 6.104 | 22.61 | 31.33 |
| | Z-Sampling | 29.31 | 1.037 | 6.118 | 22.67 | 31.30 |
| | e-CFG | 28.80 | 0.909 | 6.003 | 22.40 | 31.56 |
| **DrawBench** | CFG | 28.44 | 0.514 | 5.569 | 22.22 | 29.26 |
| | APG | 28.67 | 0.672 | 5.576 | 22.31 | 33.27 |
| | e-CFG | 29.23 | 0.770 | 5.609 | 22.36 | 33.37 |
| | CFG++ | 29.14 | 0.742 | 5.622 | 22.34 | 29.71 |
| | e-CFG | 29.05 | 0.718 | 5.600 | 22.40 | 29.73 |
| | FreeU | 28.59 | 0.567 | 5.648 | 21.86 | 28.92 |
| | e-CFG | 28.84 | 0.657 | 5.603 | 22.34 | 29.45 |
| | PAG | 28.75 | 0.530 | 5.682 | 22.21 | 29.06 |
| | e-CFG | 28.66 | 0.584 | 5.581 | 22.30 | 29.35 |
| | SAG | 28.75 | 0.599 | 5.586 | 22.30 | 32.54 |
| | e-CFG | 28.53 | 0.629 | 5.521 | 22.28 | 32.90 |
| | SEG | 28.86 | 0.533 | 5.700 | 22.14 | 28.63 |
| | e-CFG | 28.66 | 0.592 | 5.588 | 22.30 | 29.31 |
| | TDG | 28.95 | 0.723 | 5.566 | 22.39 | 29.69 |
| | e-CFG | 28.95 | 0.643 | 5.584 | 22.36 | 29.49 |
| | Z-Sampling | 29.21 | 0.724 | 5.656 | 22.37 | 29.70 |
| | e-CFG | 28.64 | 0.556 | 5.538 | 22.13 | 29.55 |
| **HPD** | CFG | 28.51 | 0.845 | 6.112 | 22.50 | 31.01 |
| | APG | 28.70 | 0.937 | 5.995 | 22.53 | 32.28 |
| | e-CFG | 29.30 | 1.032 | 5.992 | 22.61 | 32.23 |
| | CFG++ | 29.21 | 1.025 | 6.091 | 22.60 | 31.46 |
| | e-CFG | 29.05 | 0.990 | 6.103 | 22.64 | 31.39 |
| | FreeU | 28.88 | 0.929 | 6.156 | 22.11 | 30.01 |
| | e-CFG | 28.90 | 0.955 | 6.115 | 22.63 | 31.23 |
| | PAG | 28.92 | 0.863 | 6.174 | 22.47 | 30.54 |
| | e-CFG | 28.68 | 0.884 | 6.126 | 22.56 | 31.09 |
| | SAG | 28.87 | 0.884 | 6.087 | 22.55 | 30.97 |
| | e-CFG | 28.70 | 0.931 | 6.022 | 22.55 | 31.26 |
| | SEG | 28.96 | 0.875 | 6.178 | 22.35 | 30.47 |
| | e-CFG | 28.70 | 0.893 | 6.123 | 22.56 | 31.12 |
| | TDG | 28.91 | 0.972 | 6.092 | 22.61 | 31.19 |
| | e-CFG | 28.96 | 0.955 | 6.104 | 22.61 | 31.33 |
| | Z-Sampling | 29.31 | 1.037 | 6.118 | 22.67 | 31.30 |
| | e-CFG | 28.80 | 0.909 | 6.003 | 22.40 | 31.56 |

Table 6: Quantitative results of different methods. Model: Stable Diffusion-2.1 ($\omega = 5.5$).

| Dataset | Method | HPS v2 (↑) | ImageReward (↑) | AES (↑) | PickScore (↑) | CLIPScore (↑) |
|---|---|---|---|---|---|---|
| **Pick-a-Pic** | CFG | 26.25 | -0.353 | 5.658 | 20.14 | 27.08 |
| | APG | 26.03 | -0.441 | 5.675 | 20.05 | 31.97 |
| | e-CFG | 26.00 | -0.420 | 5.537 | 20.17 | 30.29 |
| | CFG++ | 27.13 | 0.045 | 5.751 | 20.43 | 27.72 |
| | e-CFG | 26.99 | -0.030 | 5.731 | 20.45 | 27.49 |
| | FreeU | 26.93 | -0.004 | 5.759 | 20.34 | 27.08 |
| | e-CFG | 26.58 | -0.251 | 5.706 | 20.30 | 27.35 |
| | PAG | 26.46 | -0.274 | 5.679 | 20.23 | 27.24 |
| | e-CFG | 26.51 | -0.258 | 5.672 | 20.24 | 27.46 |
| | SAG | 26.18 | -0.330 | 5.653 | 19.96 | 31.66 |
| | e-CFG | 26.53 | -0.240 | 5.704 | 20.25 | 32.29 |
| | Z-Sampling | 27.09 | 0.009 | 5.762 | 20.53 | 27.88 |
| | e-CFG | 26.66 | -0.124 | 5.684 | 20.20 | 28.01 |
| **DrawBench** | CFG | 27.09 | -0.218 | 5.373 | 21.13 | 27.83 |
| | APG | 26.78 | -0.304 | 5.366 | 21.02 | 31.08 |
| | e-CFG | 27.06 | -0.181 | 5.300 | 21.11 | 30.64 |
| | CFG++ | 27.96 | 0.116 | 5.474 | 21.35 | 28.68 |
| | e-CFG | 27.80 | 0.065 | 5.430 | 21.33 | 28.47 |
| | FreeU | 27.88 | 0.155 | 5.512 | 21.36 | 28.23 |
| | e-CFG | 27.55 | -0.079 | 5.405 | 21.29 | 28.04 |
| | PAG | 27.28 | -0.150 | 5.378 | 21.18 | 27.85 |
| | e-CFG | 27.26 | -0.155 | 5.387 | 21.19 | 27.99 |
| | SAG | 26.92 | -0.213 | 5.446 | 21.00 | 31.27 |
| | e-CFG | 27.17 | -0.188 | 5.361 | 21.15 | 31.30 |
| | Z-Sampling | 27.83 | 0.082 | 5.467 | 21.42 | 28.62 |
| | e-CFG | 26.96 | -0.288 | 5.377 | 21.04 | 27.98 |

Table 7: Quantitative results of different methods. Model: Stable Diffusion-3.5 ($\omega = 4.5$).

| Dataset | Method | HPS v2 (↑) | ImageReward (↑) | AES (↑) | PickScore (↑) | CLIPScore (↑) |
|---|---|---|---|---|---|---|
| **Pick-a-Pic** | CFG | 28.43 | 1.004 | 5.921 | 21.94 | 28.72 |
| | PAG | 28.49 | 0.918 | 5.994 | 21.77 | 28.19 |
| | e-CFG | 28.54 | 1.053 | 5.925 | 21.96 | 28.92 |
| | TDG | 28.56 | 0.951 | 5.890 | 21.82 | 28.57 |
| | e-CFG | 28.56 | 1.016 | 5.921 | 21.87 | 28.83 |
| | Z-Sampling | 28.30 | 0.980 | 5.853 | 21.63 | 28.74 |
| | e-CFG | 28.53 | 0.990 | 5.881 | 21.71 | 28.38 |
| **DrawBench** | CFG | 29.23 | 0.962 | 5.429 | 22.72 | 30.39 |
| | PAG | 29.25 | 0.894 | 5.531 | 22.55 | 29.70 |
| | e-CFG | 29.42 | 1.031 | 5.429 | 22.77 | 30.62 |
| | TDG | 29.42 | 0.996 | 5.435 | 22.66 | 30.50 |
| | e-CFG | 29.47 | 1.005 | 5.446 | 22.71 | 30.52 |
| | Z-Sampling | 29.32 | 0.974 | 5.434 | 22.61 | 30.17 |
| | e-CFG | 29.50 | 1.040 | 5.420 | 22.68 | 30.27 |

Table 8: Quantitative results of different methods on COCO-30K. Model: Stable Diffusion-XL ($\omega = 5.5$).

| Method | FID (↓) | HPS v2 (↑) | AES (↑) | $\omega^e$ |
|---|---|---|---|---|
| CFG | 13.56 | 28.73 | 5.558 | 5.5 |
| Z-Sampling | 17.32 | 29.56 | 5.666 | 10.81 |
| e-CFG | 14.21 | 29.22 | 5.550 | |
| CFG++ | 17.27 | 29.46 | 5.643 | 8.99 |
| e-CFG | 16.07 | 29.28 | 5.609 | |

Table 9: Quantitative results of different methods on ImageNet-50K. Model: DiT-XL/2 ($\omega = 4$).

| Method | FID ($\downarrow$) | IS ($\uparrow$) | AES ($\uparrow$) | $\omega^{\mathrm{e}}$ |
|---|---|---|---|---|
| CFG | 21.22 | 474.9 | 4.911 | 4 |
| Z-Sampling | 25.43 | 471.7 | 4.853 | 9.17 |
| e-CFG | 24.04 | 465.1 | 4.834 | |
| CFG++ | 24.33 | 480.1 | 4.950 | 8.01 |
| e-CFG | 25.55 | 480.0 | 4.791 | |

Table 10: The winning rate and its degradation of different methods on Pick-a-Pic datasets. Model: Stable Diffusion-XL ($\omega = 5.5$).

| Method | HPS v3 | OneIG |
|---|---|---|
| Pick-a-Pic | ($\eta^{\mathrm{CFG}}/\eta^{\mathrm{e\text{-}CFG}}/\Delta\eta$) | |
| PAG | 71% / 68% / 3% | 44% / 44% / 0% |
| CFG++ | 55% / 54% / 1% | 36% / 26% / 10% |
| Z-Sampling | 79% / 67% / 12% | 54% / 55% / -1% |
| APG | 44% / 39% / 5% | 24% / 26% / -2% |
| FreeU | 39% / 35% / 4% | 26% / 21% / 5% |
| TDG | 59% / 49% / 10% | 41% / 32% / 9% |
| SAG | 66% / 56% / 10% | 45% / 30% / 15% |
| SEG | 62% / 56% / 6% | 23% / 25% / -2% |

Table 11: Quantitative results of different methods on Pick-a-Pic dataset. Model: Stable Diffusion-XL ($\omega = 5.5$).

| Dataset | Model | HPS v3 ($\uparrow$) | UnifiedReward (to CFG) | UnifiedReward (to e-CFG) |
|---|---|---|---|---|
| | CFG | 6.837 | / | / |
| | PAG | 7.781 | 0.16 | 0.16 |
| | e-CFG | 7.341 | / | / |
| | CFG++ | 7.648 | 0.1 | -0.02 |
| | e-CFG | 7.612 | / | / |
| | Z-Sampling | 7.776 | 0.38 | 0.4 |
| | e-CFG | 7.203 | / | / |
| **Pick-a-Pic** | APG | 7.263 | -0.24 | -0.19 |
| | e-CFG | 7.615 | / | / |
| | FreeU | 7.038 | -0.25 | -0.32 |
| | e-CFG | 7.660 | / | / |
| | TDG | 7.493 | 0.14 | -0.03 |
| | e-CFG | 7.461 | / | / |
| | SAG | 7.432 | 0.25 | -0.03 |
| | e-CFG | 7.231 | / | / |
| | SEG | 7.534 | -0.18 | -0.14 |
| | e-CFG | 7.359 | / | / |

Table 12: Quantitative results of SD-XL on OneIG-Bench with different guidance scales.

| Guidance Scale | Alignment ($\uparrow$) | Diversity ($\uparrow$) | Reasoning ($\uparrow$) | Style ($\uparrow$) | Text ($\uparrow$) |
|---|---|---|---|---|---|
| 5.5 | .5638 | .4399 | .1932 | .3325 | .0128 |
| 6.0 | .5738 | .4357 | .1910 | .3328 | .0178 |
| 6.5 | .5771 | .4294 | .1970 | .3374 | .0170 |
| 7.0 | .5820 | .4242 | .2000 | .3420 | .0164 |
| 7.5 | .5837 | .4225 | .2076 | .3363 | .0181 |
| 8.0 | .5936 | .4169 | .2048 | .3366 | .0166 |
| 8.5 | .5917 | .4145 | .2086 | .3358 | .0182 |
| 9.0 | .5944 | .4109 | .2133 | .3411 | .0267 |
| 9.5 | .5934 | .4085 | .2154 | .3387 | .0194 |
| 10.0 | .5936 | .4043 | .2226 | .3377 | .0259 |

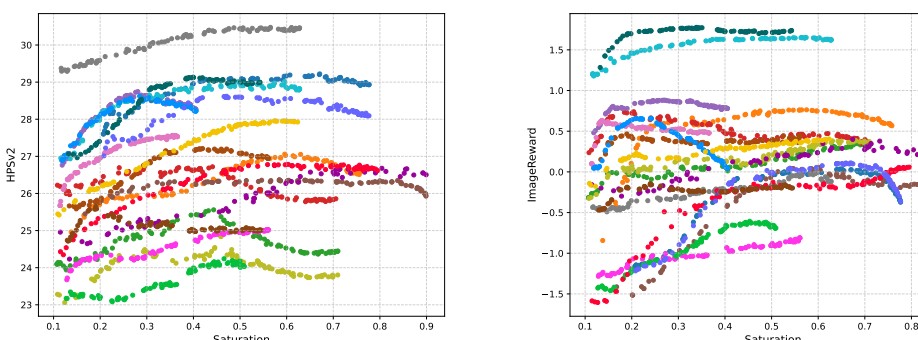

(a) Scatter map of image saturation with HPSv2    (b) Scatter map of image saturation with ImageReward

Figure 12: Reward evaluation on images with different saturation, each color represents images with different saturation. 20 images were randomly picked in MS-COCO dataset with changed saturation. Model: SD-XL.

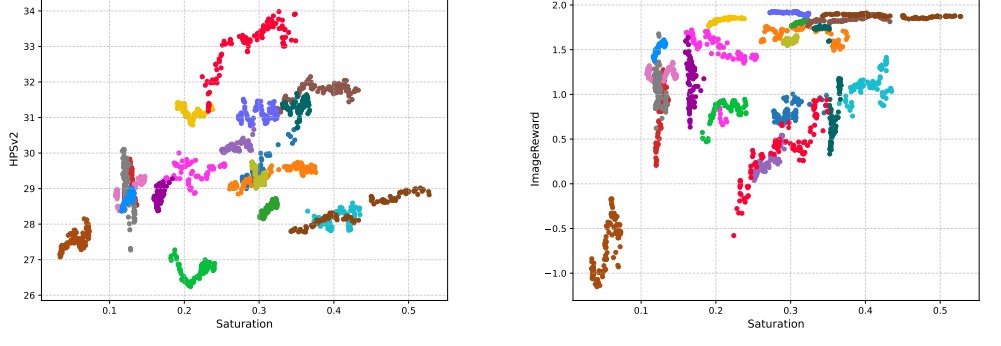

(a) Scatter map of image saturation with HPSv2    (b) Scatter map of image saturation with ImageReward

Figure 13: Reward evaluation on images with different saturation, each color represents images generated by a prompt. 20 prompts were randomly selected in Pick-a-Pic, DrawBench, and HPD, and corresponding images were generated with different guidance scale. Model: SD-XL.

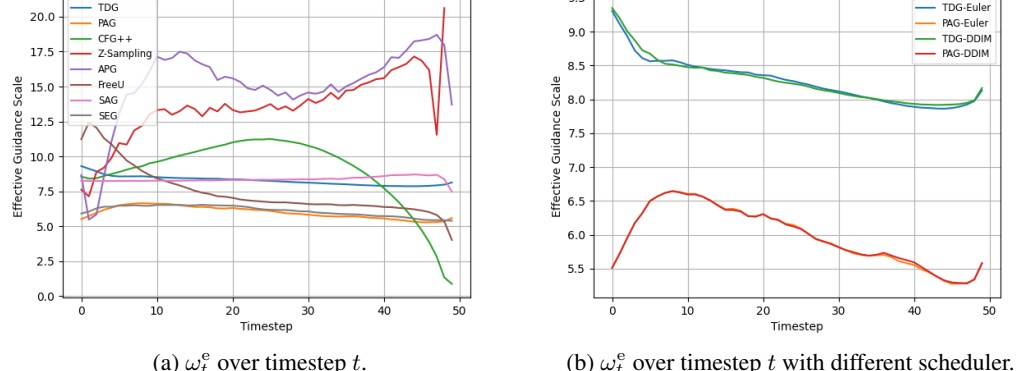

(a) $\omega_t^e$ over timestep $t$.

(b) $\omega_t^e$ over timestep $t$ with different scheduler.

Figure 14: Effective guidance scale value over timestep. Model: SD-XL. Dataset: Pick-a-Pic. Note that except Z-Sampling, CFG++, and APG, which modified the noise scheduling process, other methods use EulerDiscreteScheduler by default.

Prompt: *"A man is cooking, MineCraft Style."*

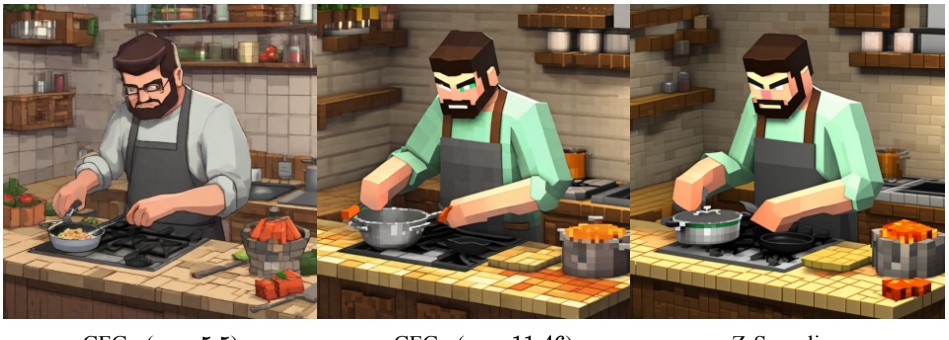

CFG  ($\omega = 5.5$)    e-CFG  ($\omega = 11.46$)    Z-Sampling

Prompt: *"A tabby cat sleeping on a wooden island in an old looking kitchen."*

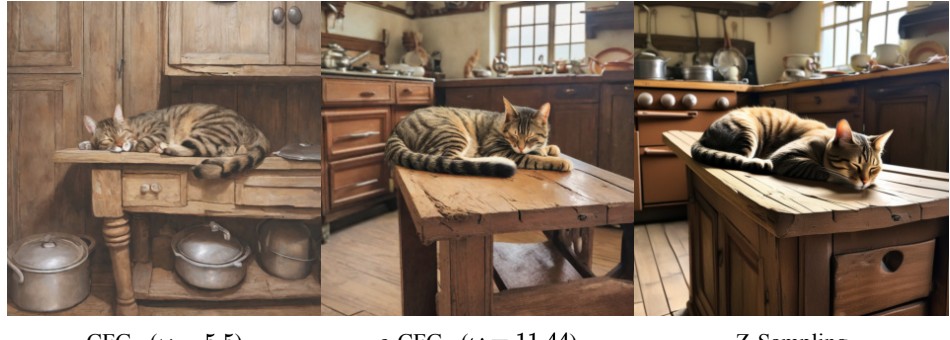

CFG  ($\omega = 5.5$)    e-CFG  ($\omega = 11.44$)    Z-Sampling

Prompt: *"The sun beyond the mountain glows, the yellow river seawards flows."*

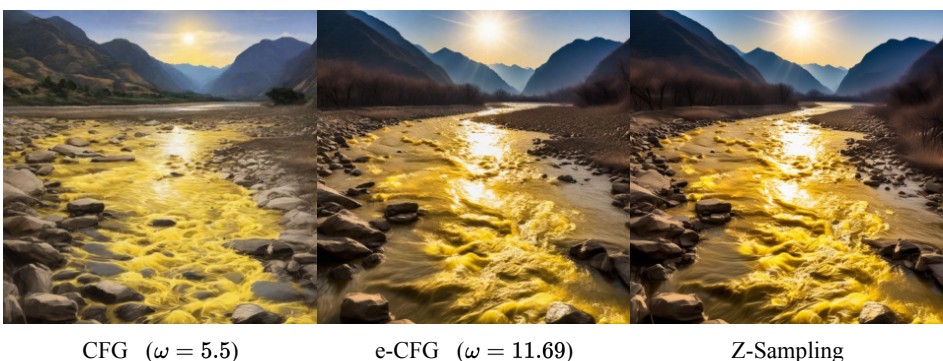

CFG  ($\omega = 5.5$)    e-CFG  ($\omega = 11.69$)    Z-Sampling

Figure 15: Qualitative Results of Z-Sampling.

Prompt: *"New York Skyline with 'Hello World' written with fireworks on the sky."*

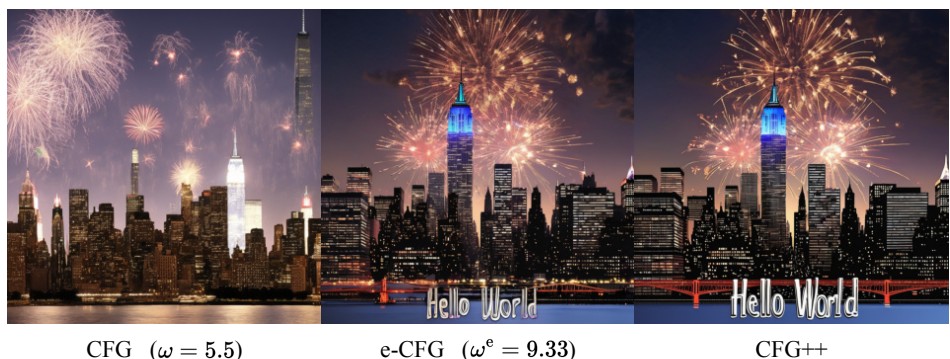

CFG  ($\omega = 5.5$)          e-CFG  ($\omega^{e} = 9.33$)          CFG++

Prompt: *"A magnifying glass over a page of a 1950s batman comic."*

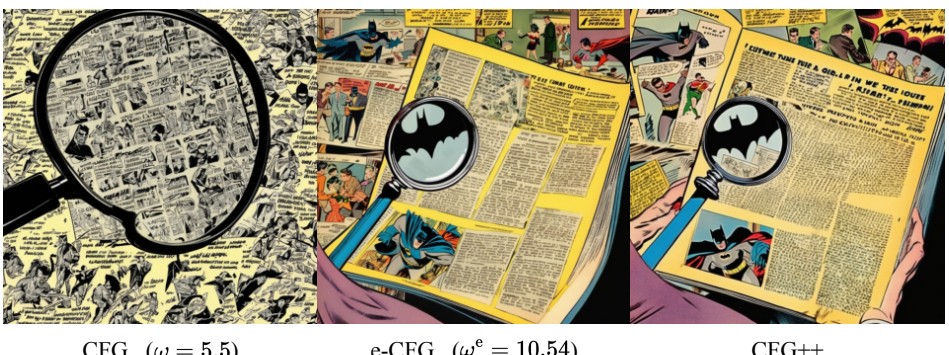

CFG  ($\omega = 5.5$)          e-CFG  ($\omega^{e} = 10.54$)          CFG++

Prompt: *"Cat wearing cowboy hat rides on corgi during sunset in the Wild West."*

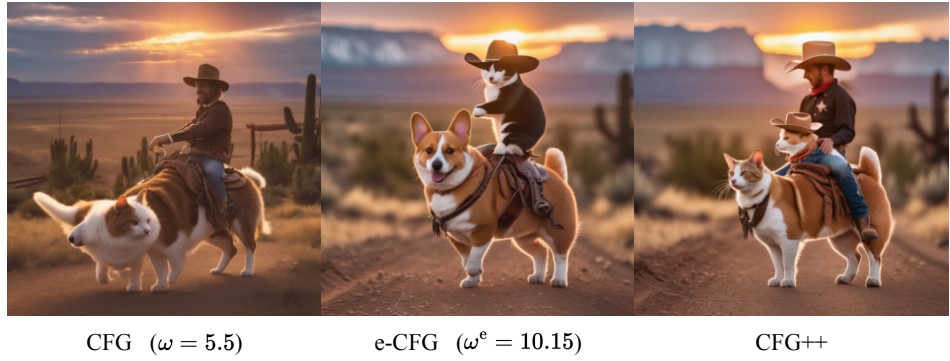

CFG  ($\omega = 5.5$)          e-CFG  ($\omega^{e} = 10.15$)          CFG++

Figure 16: Qualitative Results of CFG++.

Prompt: *"Room with a book and a white carpet."*

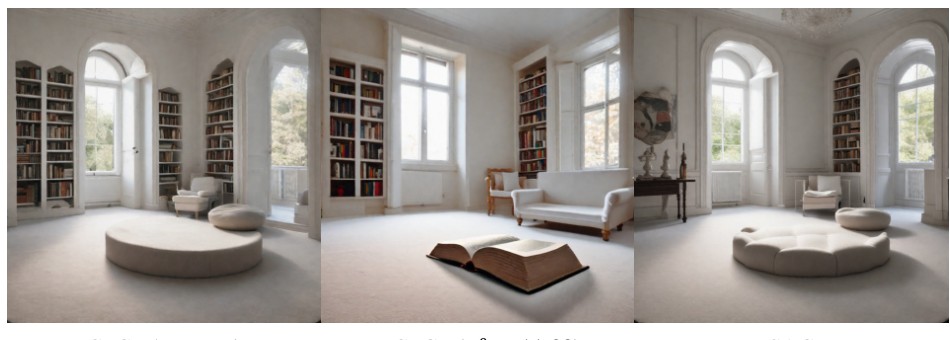

CFG $(\omega = 5.5)$        e-CFG $(\omega^{e} = 11.02)$        SAG

Prompt: *"A painting featuring a dog by artist Koyamori."*

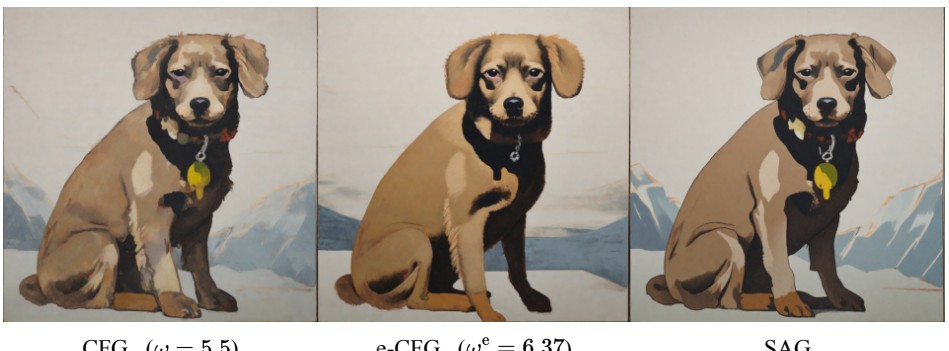

CFG $(\omega = 5.5)$        e-CFG $(\omega^{e} = 6.37)$        SAG

Prompt: *"A half-robot, half-humanoid male android, actor Liam Hemsworth, in a statue-like pose with shiny skin and a blank stare display."*

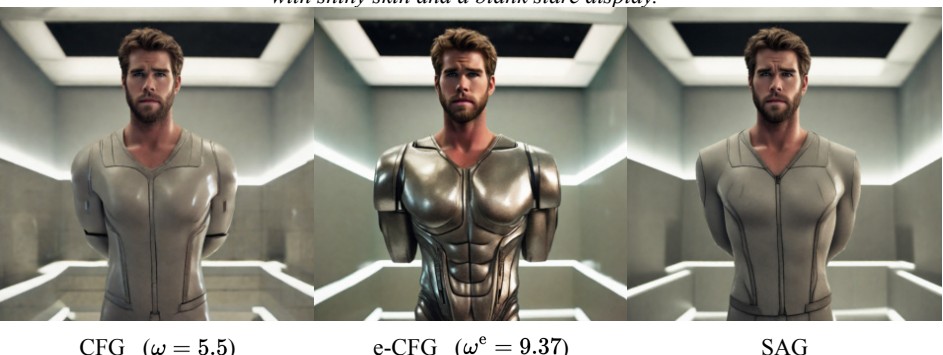

CFG $(\omega = 5.5)$        e-CFG $(\omega^{e} = 9.37)$        SAG

Figure 17: Qualitative Results of SAG.

Prompt: *"An elephant under the sea."*

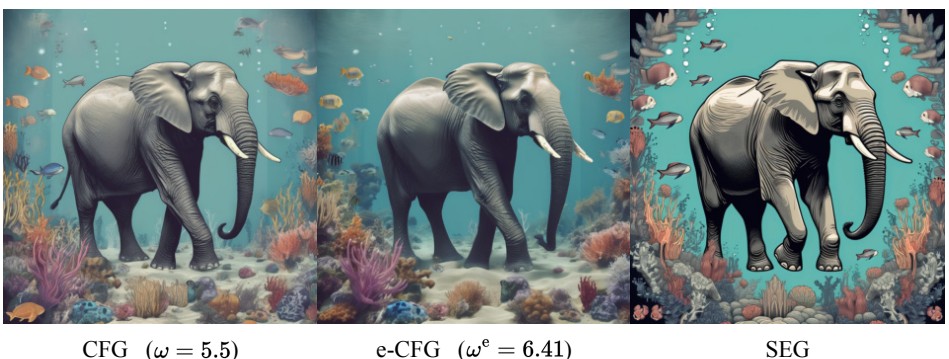

CFG  ($\omega = 5.5$)        e-CFG  ($\omega^e = 6.41$)        SEG

Prompt: *"A man and his dog riding on a bike."*

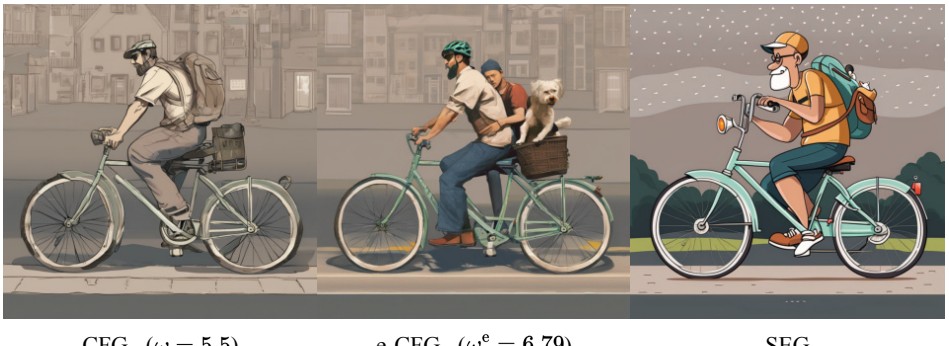

CFG  ($\omega = 5.5$)        e-CFG  ($\omega^e = 6.79$)        SEG

Prompt: *"A spanish water dog breed as Arthur Morgan from red dead redemption."*

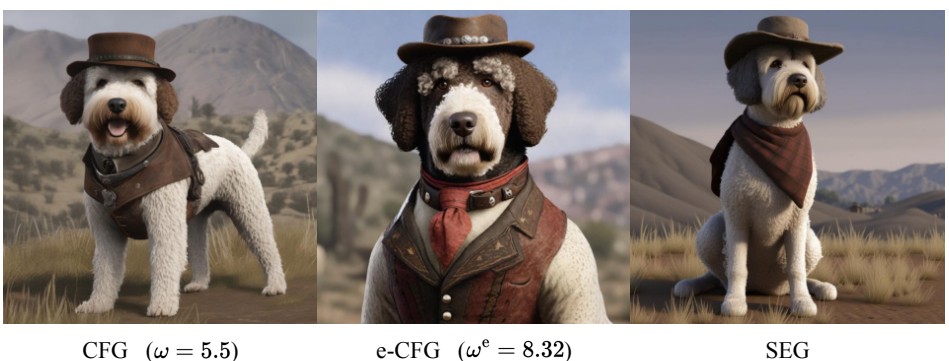

CFG  ($\omega = 5.5$)        e-CFG  ($\omega^e = 8.32$)        SEG

Figure 18: Qualitative Results of SEG.

Prompt: *"The president being abducted by aliens."*

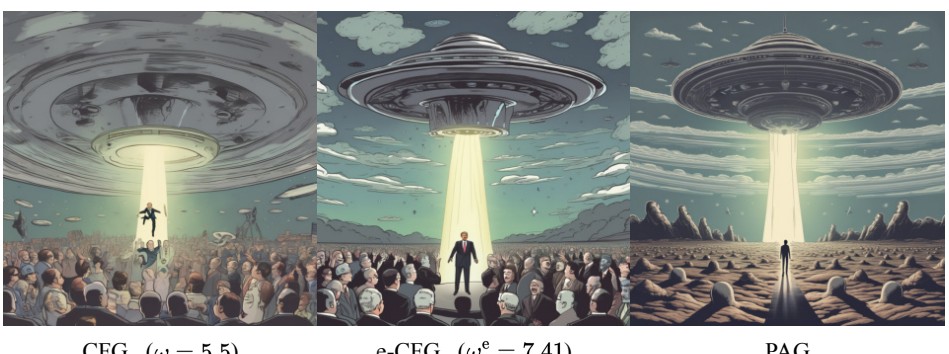

CFG  ($\omega = 5.5$)          e-CFG  ($\omega^e = 7.41$)          PAG

Prompt: *"A cramped bathroom with a sink in the corner."*

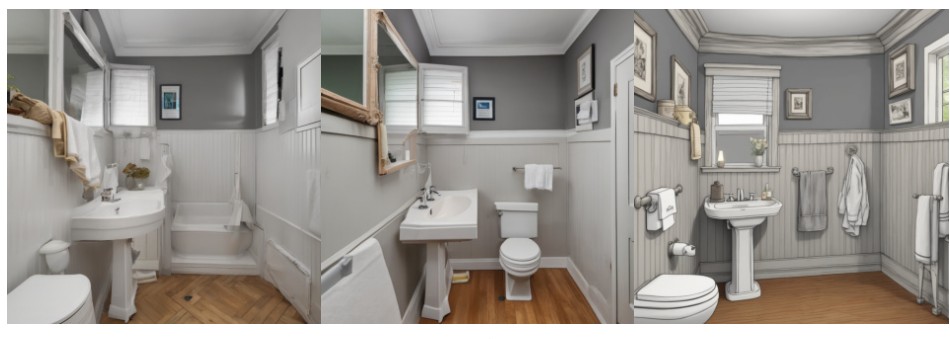

CFG  ($\omega = 5.5$)          e-CFG  ($\omega^e = 6.53$)          PAG

Prompt: *"A lynx dressed in a flight suit."*

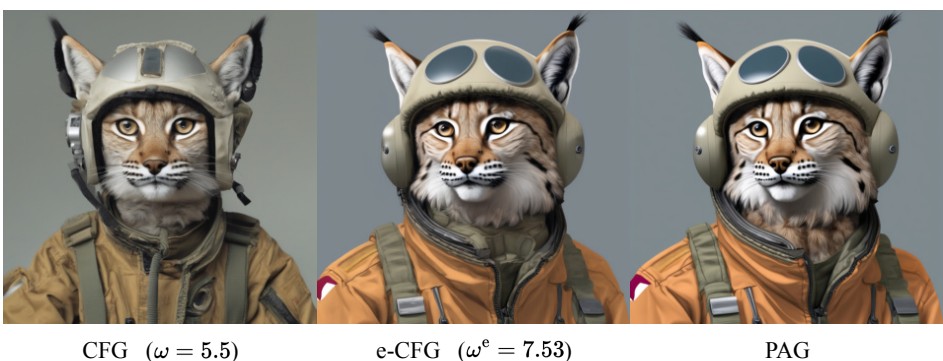

CFG  ($\omega = 5.5$)          e-CFG  ($\omega^e = 7.53$)          PAG

Figure 19: Qualitative Results of PAG.

Prompt: *"A real life photography of super mario, 8k Ultra HD."*

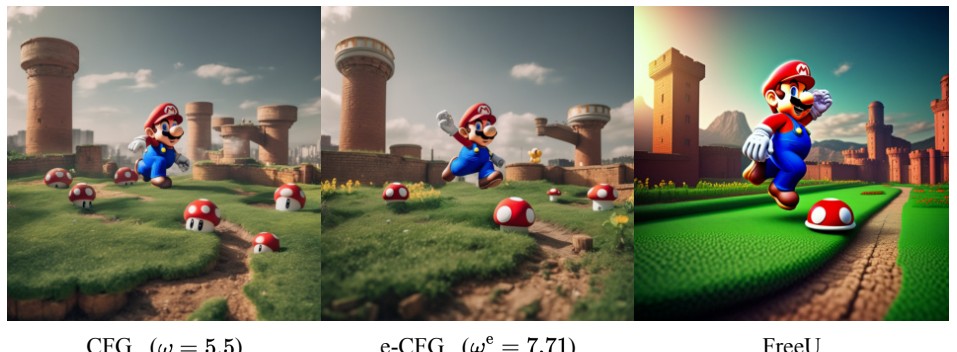

CFG ($\omega = 5.5$)      e-CFG ($\omega^e = 7.71$)      FreeU

Prompt: *"One cat and two dogs sitting on the grass."*

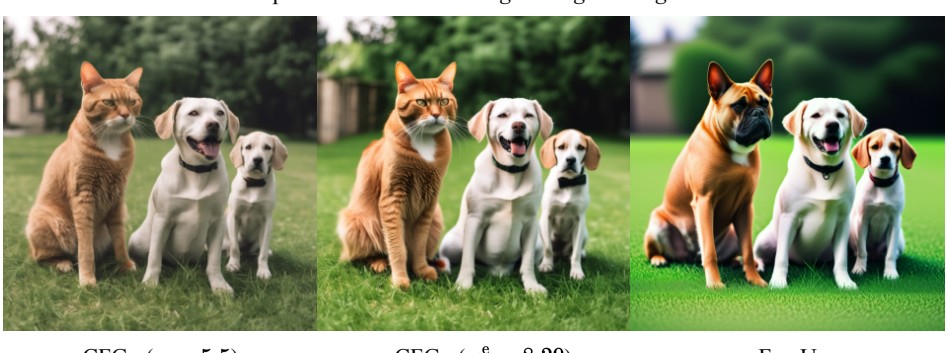

CFG ($\omega = 5.5$)      e-CFG ($\omega^e = 8.29$)      FreeU

Prompt: *"A man is standing in a field with two goats."*

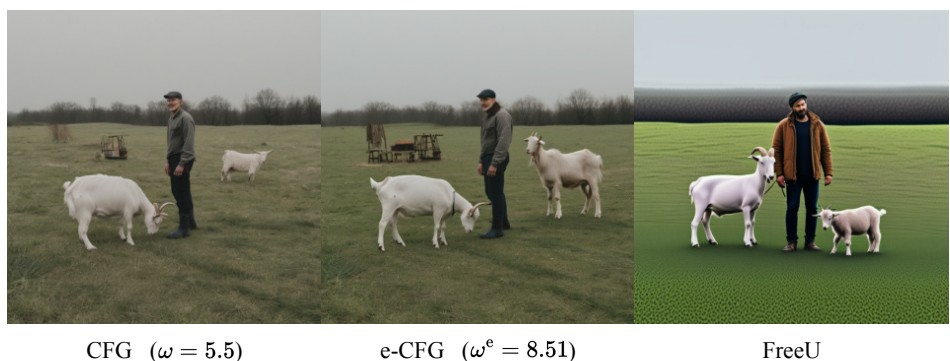

CFG ($\omega = 5.5$)      e-CFG ($\omega^e = 8.51$)      FreeU

Figure 20: Qualitative Results of FreeU.

Prompt: *"A 3D painting of a serious female sorceress in a stormy weather, with an anaglyphy effect."*

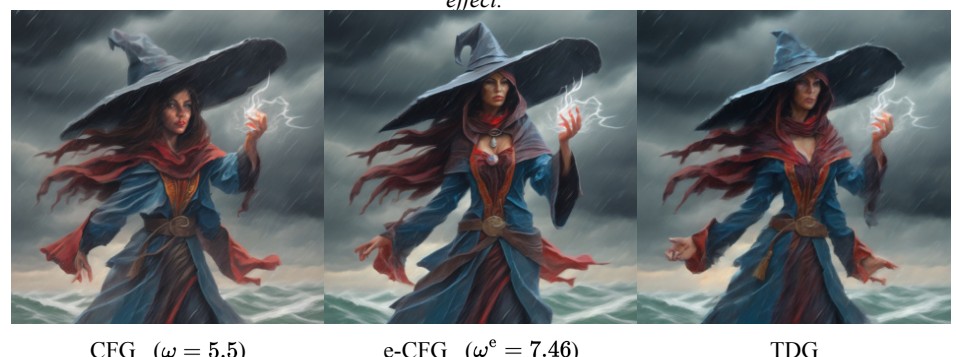

| CFG ($\omega = 5.5$) | e-CFG ($\omega^e = 7.46$) | TDG |

Prompt: *"A curly haired boy rides a skateboard down a road."*

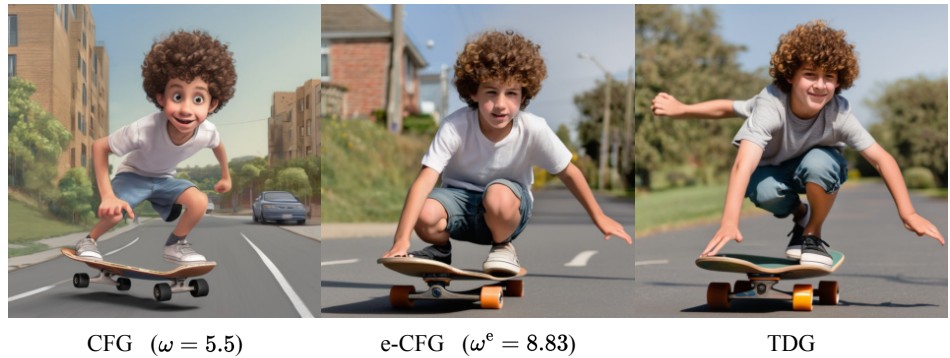

| CFG ($\omega = 5.5$) | e-CFG ($\omega^e = 8.83$) | TDG |

Prompt: *"A man and woman riding on the back of a motorcycle."*

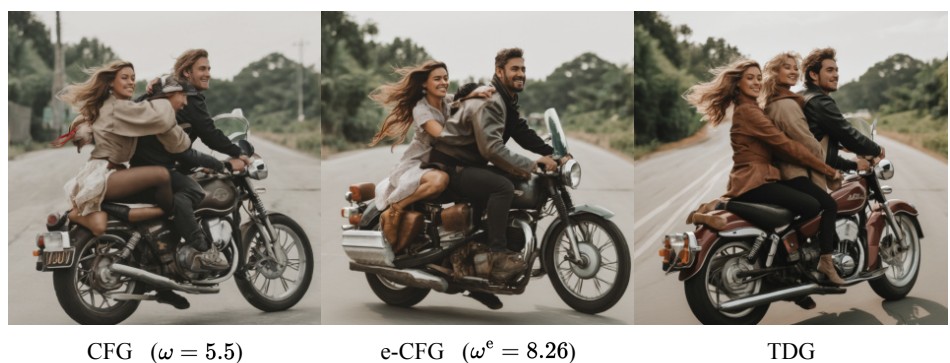

| CFG ($\omega = 5.5$) | e-CFG ($\omega^e = 8.26$) | TDG |

Figure 21: Qualitative Results of TDG.

Prompt: *"A portrait of a beautiful anime girl with pink hair wearing a white t-shirt and looking directly at the viewer."*

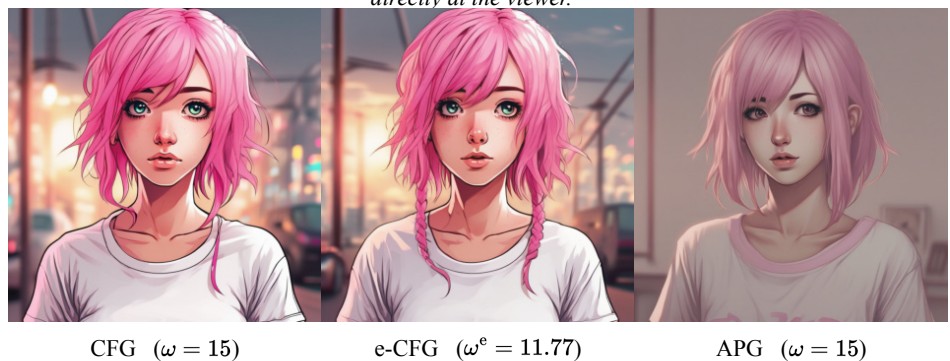

CFG  ($\omega = 15$)  e-CFG  ($\omega^e = 11.77$)  APG  ($\omega = 15$)

Prompt: *"Lego Arnold Schwarzenegger."*

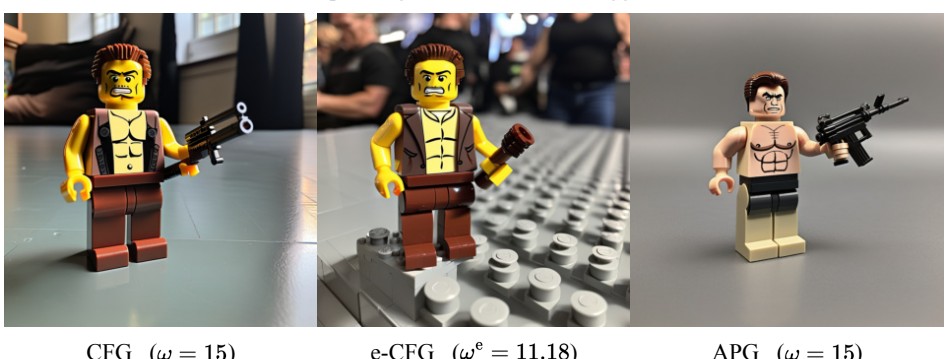

CFG  ($\omega = 15$)  e-CFG  ($\omega^e = 11.18$)  APG  ($\omega = 15$)

Prompt: *"An image of a fantastical city floating in the clouds."*

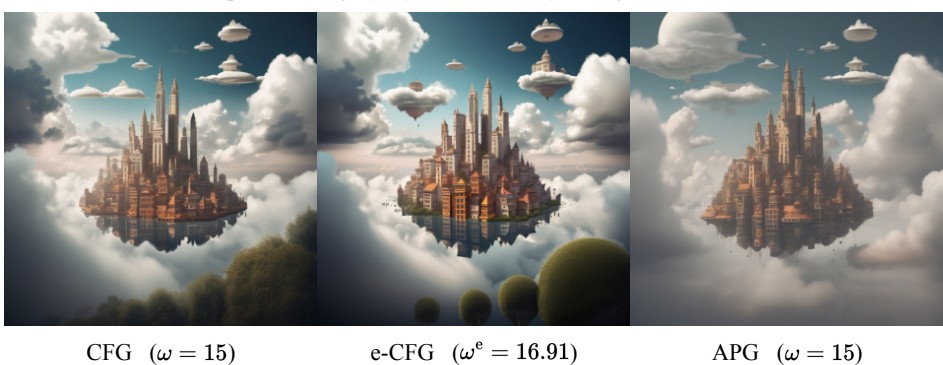

CFG  ($\omega = 15$)  e-CFG  ($\omega^e = 16.91$)  APG  ($\omega = 15$)

Figure 22: Qualitative Results of APG.

