# OpenReview forum: "Guidance Matters: Rethinking the Evaluation Pitfall for Text-to-Image Generation"
_ICLR.cc/2026/Conference — ICLR 2026 Poster_

### Official Review · Reviewer_5uJX · 2025-10-16

**Soundness:** 3
**Presentation:** 3
**Contribution:** 2
**Rating:** 6
**Confidence:** 4

**Summary:**

This paper identifies and analyzes a critical yet overlooked evaluation pitfall in diffusion model research-specifically, the tendency of human preference models to favor high CFG scales due to their implicit bias toward color saturation and semantic alignment.
To address this, the authors propose a Guidance-Aware Evaluation (GA-Eval) framework, which introduces an“effective guidance scale”to disentangle the actual improvement from the simple effect of scale amplification. They further design a synthetic baseline called Transcendent Diffusion Guidance (TDG), demonstrating that it can“fool”conventional evaluation metrics without real quality improvement.
Through experiments across multiple diffusion backbones (SD-XL, SD-2.1, SD-3.5, DiT-XL/2) and datasets (Pick-a-Pic, HPD, DrawBench, GenEval, COCO-30K), they argue that many recent guidance methods achieve their gains mainly by exploiting large-scale biases rather than genuine improvements.

**Strengths:**

①Timely and meaningful contribution. The paper addresses an important meta-evaluation problem in the rapidly expanding field of text-to-image diffusion models. The argument that large guidance scales bias reward-based evaluation metrics is well-motivated and empirically observable.
②Novel evaluation framework (GA-Eval). Introducing the effective guidance scale and decomposing noise updates into orthogonal and parallel components provides an interesting analytical perspective for isolating true guidance effects.
③Strong experimental coverage. The authors compare eight diffusion guidance methods on four major backbones and five evaluation metrics, offering a relatively broad experimental scope.

**Weaknesses:**

(A) Theoretical ambiguity in ωₑ definition
The derivation of effective guidance scale (Eq.4-7) assumes linearity between the unconditional and conditional noise terms. However, in practical schedulers (e.g., DDIM, ODE-based), the latent evolution is nonlinear.

(B) Limited validation of GA-Eval fairness
The paper claims GA-Eval enables“fair comparison,”but provides no user study or human evaluation to verify that its results better correlate with actual human perception. Without human validation, GA-Eval could itself introduce a new bias.

(C) TDG lacks principled formulation
TDG randomly masks text tokens to produce a“weakened prompt,”then combines multiple noise terms (Eq. 13).
This method appears heuristic and lacks a theoretical foundation for why such hyperplane mixing would improve quality.
The scaling factor∥ϵ_cond−ϵ_uncond∥/∥ϵ_cond−ϵ_weak∥seems ad hoc and could destabilize training or sampling.

(D) Incomplete quantitative analysis of metric bias
The claim that reward models favor high-saturation images is intuitively correct but not statistically demonstrated. No quantitative correlation between color saturation and metric scores is provided. Figure 3 illustrates trends, but lacks regression or significance testing.

**Questions:**

1.How is the linearity assumption in the derivation of ωₑ justified, given the nonlinear nature of DDIM/ODE schedulers?
2.Have you analyzed ωₑ stability across timesteps or different sampling schedulers?
3.How is GA-Eval’s “fairness” verified without human evaluation? Any correlation with human judgments?
4.Could GA-Eval itself introduce bias (e.g., penalizing certain visual styles)? What is the theoretical motivation for the TDG noise-mixing rule and its scaling factor?
5.How sensitive is TDG to masking ratio or weakened prompt settings? Can you provide statistical evidence (e.g., correlation or regression) showing that reward models favor high-saturation images?

---

> ### Author Response · Authors · 2025-11-21
>
> We'd like to thank you for the comprehensive comments and positive perceptions, and we now address each of the given concerns:
>
> **WA:** Theoretical ambiguity about linearity in $\omega^\mathrm{e}$ definition.
>
> A: Thanks for the comment. We want to clarify that we didn't made any assumption in the definition of $\omega^\mathrm{e}$.  in order to test how many components in other methods $\tilde{\epsilon}_t^*$ are parallel to $\Delta \epsilon $, we decomposed its components. A vector can always be decomposed into two orthogonal vectors satisfies $\epsilon_t^{\perp} \perp \epsilon_t^{\parallel }$, and is independent of the nonlinear properties of latent space evolution. Besides, APG also uses a similar method to calculate parallel/vertical components on the latent, suppressing the saturation of images at high guidance scales. In general, we didn't assume the linearity in the decomposition operation, and has nothing to do with non-linearity of latent evolution in different schedulers.
>
> **WB:** Lack of human evaluation impairs GA-Eval's fairness.
>
> A: Thank you for pointing out this key point. We would like to emphasize that the core purpose GA-Eval framework is not to directly replace or precisely mimic human preference assessments, but rather to serve as a diagnostic tool for revealing the prevalent evaluation bias in the existing evaluation paradigm. In fact, slightly increasing the guidance scale will increase the saturation of generated image [1], making it more in line with human preferences [2], and this trend is also implicit in reward's training dataset.
>
> **WC:**  TDG lacks principled formulation.
>
> A: Thanks for your concern, we have demonstrated in **Q4 b)**.
>
> **WD:** No quantitative correlation between color saturation and metric scores is provided.
>
> A: Thanks for your suggestion. We have illustraed this in **W1** of reviewer **5pnH**, as well as in Fig.12 and Fig.13 in the revised manuscript. In general, the relation of saturation and metric score are statistically correlated.
>
> **Q1:** How is the linearity assumption in the derivation of ωₑ justified, given the nonlinear nature of DDIM/ODE schedulers?
>
> A: We have demonstrated that in **WA**. There is no linearity assumption in the derivation of ωₑ.
>
> **Q2:** Have you analyzed ωₑ stability across timesteps or different sampling schedulers?
>
> A: Thanks for the question. We draw the curve of effective guidance scale over timestep. But due to some methods, like Z-Sampling or CFG++ need to use their own scheduler, we only test different scheduler on TDG and PAG for example. The results can be found in Fig.14 in the revised manuscript. Most of methods have average $\omega^\mathrm{e}$ larger than baseline 5.5, and their trends are different from one to another. In Fig.14(b), it can be found that different scheduler didn't affect the effective guidance scale.

---

> ### Author Response · Authors · 2025-11-21
>
> **Q3:** How is GA-Eval’s “fairness” verified without human evaluation?
>
> A: We have demonstrated that in **WB**.
>
> **Q4:** a) Could GA-Eval itself introduce bias (e.g., penalizing certain visual styles)? b) What is the theoretical motivation for the TDG noise-mixing rule and its scaling factor?
>
> A:
> a) This problem does exist. APG achieves relatively normal image saturation at a larger guidance scale (w=15). However, after GA-Eval evaluations, APG was treated unfairly: although APG did not utilize the evaluation pitfall like other methods, the calculated effective guidance scale was still very high, resulting in weaker metric scores in GA-Eval compared to CFG using the effective guidance scale.
>
> b)
>
> | g\β | 2.00 | 2.20 | 2.40 | 2.60 | 2.80 | 3.00 |
> |:---:|:---:|:---:|:---:|:---:|:---:|:---:|
> | 1.00 | 0.67,0.33 | 0.69,0.31 | 0.71,0.29 | 0.72,0.28 | 0.74,0.26 | 0.75,0.25 |
> | 1.20 | 0.80,0.40 | 0.82,0.37 | 0.85,0.35 | 0.87,0.33 | 0.88,0.32 | 0.90,0.30 |
> | 1.40 | 0.93,0.47 | 0.96,0.44 | 0.99,0.41 | 1.01,0.39 | 1.03,0.37 | 1.05,0.35 |
> | 1.60 | 1.07,0.53 | 1.10,0.50 | 1.13,0.47 | 1.16,0.44 | 1.18,0.42 | 1.20,0.40 |
> | 1.80 | 1.20,0.60 | 1.24,0.56 | 1.27,0.53 | 1.30,0.50 | 1.33,0.47 | 1.35,0.45 |
> | 2.00 | 1.33,0.67 | 1.38,0.62 | 1.41,0.59 | 1.44,0.56 | 1.47,0.53 | 1.50,0.50 |
>
> We set guidnce scale factor (g) and balance scale factor (β) to control the coefficients of ($\epsilon^{\mathrm{cond}}_t-\epsilon^{\mathrm{uncond}}_t$) and ($\epsilon^{\mathrm{cond}}_t-\epsilon^{\mathrm{weak}}_t$) , and each element in the table above corresponds to these two coefficients. When g is fixed and β increasing, the ratio of the coefficient of ($\epsilon^{\mathrm{cond}}_t-\epsilon^{\mathrm{uncond}}_t$) to the coefficient of ($\epsilon^{\mathrm{cond}}_t-\epsilon^{\mathrm{weak}}_t$) will gradually increase, that is, ($\epsilon^{\mathrm{cond}}_t-\epsilon^{\mathrm{weak}}_t$) will decrease compared to ($\epsilon^{\mathrm{cond}}_t-\epsilon^{\mathrm{uncond}}_t$). When b is fixed and g is increasing, the ratio of the two coefficients remains unchanged. In addition, the introduction of $\frac{\left \| \epsilon^{\mathrm{cond}}_t - \epsilon^{\mathrm{uncond}}_t \right \|}{\left \| \epsilon^{\mathrm{cond}}_t - \epsilon^{\mathrm{weak}}_t \right \|}$ is also to ensure that the modulus length of ($\epsilon^{\mathrm{cond}}_t-\epsilon^{\mathrm{uncond}}_t$) is consistent with that of ($\epsilon^{\mathrm{cond}}_t-\epsilon^{\mathrm{weak}}_t$). In fact, we determined the optimal hyperparameters of TDG by performing grid search on g and β on HPSv2.
>
> Due to this operation indirectly amplifying the guidance scale, TDG has gained an advantage over the original CFG in HPSv2. Can simply increasing the guidance scale achieve comparable results? Did other methods also exploit this evaluation bias? This inspired us to design the GA-Eval framework.
>
> **Q5:** a) How sensitive is TDG to masking ratio or weakened prompt settings? b) Can you provide statistical evidence (e.g., correlation or regression) showing that reward models favor high-saturation images?
>
> A: a) We test TDG's performance with different mask ratio. There is no significant performance degradation with mask ratio not greater than 0.6. The results are presented in Fig.10 in the revised manuscript.
>
> b) We have illustraed this in **W1** of reviewer **5pnH**, as well as in Fig.12 and Fig.13 in the revised manuscript. In general, the relation of saturation and metric score are statistically correlated.
>
> [1] Sadat, Seyedmorteza, Otmar Hilliges, and Romann M. Weber. "Eliminating oversaturation and artifacts of high guidance scales in diffusion models." The Thirteenth International Conference on Learning Representations. 2024.
>
> [2] Lin, Chenyang, Sabrina Mottaghi, and Ladan Shams. "The effects of color and saturation on the enjoyment of real-life images." Psychonomic bulletin & review 31.1 (2024): 361-372.

---

> ### Author Response · Authors · 2025-11-28
>
> Dear Reviewer:
>
> We hope this message finds you well. As the discussion period is nearing its end, we hope that we have addressed all your concerns satisfactorily. If there are any additional points or feedback you'd like us to consider, please let us know. Your insights are invaluable to us, and we're eager to address any remaining issues to improve our work.
>
> Thank you for your time and effort in reviewing our paper.

---

### Official Review · Reviewer_5pnH · 2025-10-31

**Soundness:** 3
**Presentation:** 3
**Contribution:** 3
**Rating:** 6
**Confidence:** 2

**Summary:**

This paper critically examines the evaluation practices in recent text-to-image (T2I) diffusion guidance methods. The authors identify a pervasive evaluation pitfall: human-preference-based metrics exhibit a strong bias toward images generated with large classifier-free guidance (CFG) scales, even when such images suffer from visual artifacts like oversaturation or loss of fidelity. This bias leads to inflated performance claims for many newly proposed guidance techniques.

To address this, the authors make four key contributions: (1) revealing the pitfall; (2) proposing GA-Eval; (3) Designing TDG: A misleading Transcendent Diffusion GuidanceThe work calls for a paradigm shift in T2I evaluation, urging the community to disentangle true algorithmic innovation from artifacts of biased metrics.

**Strengths:**

The paper is highly original in both problem framing and methodology. While prior works have proposed new guidance strategies, this is the first to systematically diagnose and quantify a systematic bias in human-preference metrics tied to CFG scale. The technical execution is rigorous. The derivation of the effective guidance scale is mathematically sound and generalizable across sampling algorithms (including those that modify latents rather than noise directly). Experiments are extensive: multiple models, datasets, and metrics are tested, and results are consistently interpreted through the lens of GA-Eval. The inclusion of GenEval for fine-grained semantic alignment and COCO/ImageNet for distributional metrics (FID/IS) further strengthens validity.

**Weaknesses:**

1. Authors could provide more details on how HPS v2, ImageReward, etc., are biased toward high-saturation/high-alignment images
2. While GA-Eval is a diagnostic tool, the paper offers limited direction on how to design guidance methods that truly improve generation beyond CFG scaling.
3. Some recent text-to-image works might need examination in this paper [1,2]. It is intriguing to investigate whether these recent generative reward model are still biased.

[1] Unified Multimodal Chain-of-Thought Reward Model through Reinforcement Fine-Tuning
[2] T2I-Eval-R1: Reinforcement Learning-Driven Reasoning for Interpretable Text-to-Image Evaluation

**Questions:**

No question

---

> ### Author Response · Authors · 2025-11-21
>
> We wish to thank the reviewer for the helpful comments and for finding our work novel with detailed extensive evaluations. Now we address each of the given concerns:
>
> **W1:** More details on how HPS v2, ImageReward, etc., are biased toward high-saturation/high-alignment images.
>
> Thanks for the advice. We further valid this phenomenon through 2 aspects: 1) Real images with different saturation; 2) Generated images with different guidance scale. We also use Spearman' test to valid the correlations between the image saturation and metric score, the alpha value was set to 0.05.
>
> 1) For real images, we randomly selected 20 Ground-Truth images from MS-COCO dataset, and changed their saturation with different levels. The scatter map are presented in Fig. 12 in the revised manuscript. Most prompts have metric score positively correlated with image saturation. In HPSv2, Significant positive correlations: 14, Positive correlation not significant: 1, No significant positive correlation: 5. In ImageReward, Significant positive correlations: 14, Positive correlation not significant: 2, No significant positive correlation: 4.
>
> 2) For generated images, we randomly select 20 prompts from Pick-a-Pic, DrawBench, and HPD. Then generated then with different guidance scales. The scatter map are presented in Fig. 13 in the revised manuscript. And the problem still remains: In HPSv2, Significant positive correlations: 15, Positive correlation not significant: 2, No significant positive correlation: 3. In ImageReward, Significant positive correlations: 12, Positive correlation not significant: 1, No significant positive correlation: 7.
>
> **W2:** Limited contribution on how to design guidance methods truly improve generation.
>
> Thank you to the reviewer for raising this question. We agree that this article does not provide specific design directions to surpass CFG scaling. However, the core value of GA-Eval is a diagnostic tool to test whether other methods have exploited evaluation bias. In addition, GA-Eval provides a perspective for decomposing guidance signals into orthogonal and parallel components, providing a rough direction for future guidance method design. Our viewpoint is not simply to judge which component is more useful, but in designing guidance methods, if performance improvement is achieved, further testing is needed to determine whether this is due to the parallel component.
>
> **W3:** Recent metrics and benchmarks need examinations.
>
> Thank you for pointing this out. We have presented the results of UnifiedReward in Table 10-12 in the revised manuscript, and also in the **W2** of Reviewer **2V4Q**. Due to lack of open-source code of T2I-Eval-R1, we can't conduct experiments on it. But from the results of other recent metrics and benchmarks, we can still found the prevalent existence of such evaluation bias.

---

> > ### Comment · Reviewer_5pnH · 2025-11-23
> >
> > Thank you for the detailed reply. I feel that your response partially addressed my concerns, so I’ve decided to keep the current score.

---

### Official Review · Reviewer_2V4Q · 2025-10-31

**Soundness:** 3
**Presentation:** 2
**Contribution:** 3
**Rating:** 6
**Confidence:** 4

**Summary:**

This paper critically analyzes recent progress in diffusion guidance for text-to-image generation, focusing on four key contributions. It identifies an evaluation pitfall where human preference models favor larger guidance scales, affecting image quality. The authors propose a guidance-aware evaluation (GA-Eval) framework for fair comparisons between methods and classifier-free guidance (CFG). They introduce the Transcendent Diffusion Guidance (TDG) method, which boosts preference scores but is ineffective in practice. Experiments show that increasing CFG scales can compete with most methods, though all suffer from decreased winning rates compared to standard CFG. The work urges reconsideration of the evaluation paradigm and future directions in this field.

**Strengths:**

1. This paper emphasizes a significant evaluation flaw in which standard human preference models show a strong bias toward larger guidance scales. This is crucial for the field as it encourages a reevaluation of the contributions of each classifier-free guidance (CFG) method.
2. The proposed guidance-aware evaluation (GA-Eval) framework is both reasonable and robust. It will assist researchers in accurately assessing diffusion guidance methods.
3. The introduced Transcendent Diffusion Guidance (TDG) method replicates the creation of weak conditions found in other methods during the sampling process. It highlights the effectiveness of GA-Eval and enhances the overall persuasiveness of the paper.

**Weaknesses:**

1. While GA-Eval is mathematically sound, it requires user studies to demonstrate its alignment with human assessments.
2. This paper identifies an evaluation flaw and proposes a guidance-aware evaluation framework. Another approach to addressing this issue is to enhance the reward model and testing benchmarks. Recent reward models, like HPSv3, may rely on vision-language models, and recent benchmarks, such as OneIG, also utilize VLMs to evaluate generated images. These new evaluation tools could potentially diminish the significance of this paper.

**Questions:**

None

---

> ### Author Response · Authors · 2025-11-21
>
> We thank the reviewer for the positive reception of our paper and for recognizing its strengths. We also thank the reviewer for the opportunity to clarify several points.
>
> **W1:** GA-Eval requires user studies to demonstrate its alignment with human assessments.
>
> A: Thank the reviewer for raising an important point about the consistency of human assessments. We would like to emphasize that the core purpose GA-Eval framework is not to directly replace or precisely mimic human preference assessments, but rather to serve as a diagnostic tool for revealing the prevalent evaluation bias in the existing evaluation paradigm.
>
> In fact, humans themselves may also exhibit subtle and inherent perceptual biases. For instance, as indicated by some visual psychology studies [1], humans do indeed exhibit a preference for images with slightly enhanced saturation or contrast to some extent, and a larger guidance scale often indirectly leads to such high-saturation results. This, in turn, reinforces our argument: the combination of evaluation pitfalls and subtle human preferences makes the existing methods results becoming misleading. Therefore, GA-Eval lies in providing a Guidance-Aware benchmark to promote the field to rethink evaluation paradigms and future research directions.
>
> **W2:** Lack of recent reward models based on VLM, such as HPSv3 and OneIG.
>
> A: Thanks for your suggestions. We now test some recent VLM-based reward models. HPSv3 still hold this evaluation pitfall, with most methods suffer winning rate degradation. Meanwhile, due to OneIG-Bench require to use theirs prompts to evaluate, we only conduct experiments on SD-XL with different guidance scales. The Alignment score, Reasoning score, and Text score are increasing with guidance scale. But Diversity score decreases on the contrary, this also meet the discovery in [2]. These results are also presented in Table 10-12 in the revised manuscript.
>
>
> The winning rate and its degradation of different methods on Pick-a-Pic datasets. Model: Stable Diffusion-XL ($\omega=5.5$).
> | Method | HPS v3 ($\eta^{\text{CFG}} / \eta^{\text{e-CFG}}/\Delta\eta$) | UnifiedReward ($\eta^{\text{CFG}} / \eta^{\text{e-CFG}}/\Delta\eta$) |
> | :--- | :--- | :--- |
> | **PAG** | 71\% / 68\% / 3\% | 44\% / 44\% / 0\% |
> | **CFG++** | 55\% / 54\% / 1\% | 36\% / 26\% / 10\% |
> | **Z-Sampling** | 79\% / 67\% / 12\% | 54\% / 55\% / -1\% |
> | **APG** | 44\% / 39\% / 5\% | 24\% / 26\% / -2\% |
> | **FreeU** | 39\% / 35\% / 4\% | 26\% / 21\% / 5\% |
> | **TDG** | 59\% / 49\% / 10\% | 41\% / 32\% / 9\% |
> | **SAG** | 66\% / 56\% / 10\% | 45\% / 30\% / 15\% |
> | **SEG** | 62\% / 56\% / 6\% | 23\% / 25\% / -2\% |
>
>
> Quantitative results of SD-XL on OneIG-Bench with different guidance scales.
> | Guidance Scale | Alignment | Diversity | Reasoning | Style | Text |
> | :---: | :---: | :---: | :---: | :---: | :---: |
> | 5.5 | .5638 | .4399 | .1932 | .3325 | .0128 |
> | 6.0 | .5738 | .4357 | .1910 | .3328 | .0178 |
> | 6.5 | .5771 | .4294 | .1970 | .3374 | .0170 |
> | 7.0 | .5820 | .4242 | .2000 | .3420 | .0164 |
> | 7.5 | .5837 | .4225 | .2076 | .3363 | .0181 |
> | 8.0 | .5936 | .4169 | .2048 | .3366 | .0166 |
> | 8.5 | .5917 | .4145 | .2086 | .3358 | .0182 |
> | 9.0 | .5944 | .4109 | .2133 | .3411 | .0267 |
> | 9.5 | .5934 | .4085 | .2154 | .3387 | .0194 |
> | 10.0 | .5936 | .4043 | .2226 | .3377 | .0259 |
>
> [1] Lin, Chenyang, Sabrina Mottaghi, and Ladan Shams. "The effects of color and saturation on the enjoyment of real-life images." Psychonomic bulletin & review 31.1 (2024): 361-372.
>
> [2] Karras, Tero, et al. "Guiding a diffusion model with a bad version of itself." Advances in Neural Information Processing Systems 37 (2024): 52996-53021.

---

> ### Author Response · Authors · 2025-11-28
>
> Dear Reviewer:
>
> We hope this message finds you well. As the discussion period is nearing its end, we hope that we have addressed all your concerns satisfactorily. If there are any additional points or feedback you'd like us to consider, please let us know. Your insights are invaluable to us, and we're eager to address any remaining issues to improve our work.
>
> Thank you for your time and effort in reviewing our paper.

---

### Official Review · Reviewer_JH4a · 2025-10-31

**Soundness:** 1
**Presentation:** 1
**Contribution:** 2
**Rating:** 2
**Confidence:** 4

**Summary:**

This paper critically reexamines recent diffusion guidance methods beyond Classifier-Free Guidance (CFG) and exposes an evaluation bias: human preference models such as HPSv2 and ImageReward tend to favor images with higher CFG scales, which often leads to overly saturated or artifact-prone generations. To address this, the authors propose GA-Eval, a guidance-aware evaluation framework that calibrates guidance scale effects to ensure fair comparisons across methods. They further introduce Transcendent Diffusion Guidance (TDG), a method that appears to outperform CFG under biased evaluations but in reality does not improve generation quality.
The experiments show that, when accounting for this bias, simply increasing CFG scale can match or outperform most recent guidance methods, urging the community to rethink evaluation protocols for diffusion guidance.

**Strengths:**

The paper makes an insightful observation that commonly used human preference models, such as HPSv2 and ImageReward, exhibit a strong bias toward images generated with larger CFG scales. This bias frequently leads to higher preference scores for oversaturated or artifact-prone images, revealing a critical weakness in current evaluation protocols for diffusion guidance methods.

**Weaknesses:**

1. Writing clarity: The overall writing could be improved to enhance readability and make the key ideas easier to follow. Some sections, particularly the methodological descriptions, are difficult to interpret without additional context or intuitive explanations.

2. Equations (5)–(7): The derivations and computation steps for Equations (5) through (7) are not clearly explained. It would be helpful if the authors explicitly detailed how these equations are obtained, what intermediate steps are omitted, and what assumptions or approximations are involved in their formulation.

3. Motivation of Transcendent Diffusion Guidance (TDG): The introduction of the proposed TDG method lacks sufficient motivation and a clear conceptual link to the findings presented earlier. It remains unclear why this method should be effective or how it concretely relates to the identified evaluation bias. Providing a stronger theoretical or intuitive justification for TDG would make the contribution more convincing.

**Questions:**

1. Could the authors clarify how the components that are orthogonal and parallel to $\Delta \epsilon$ are computed in Equation 5-7?     Specifically, what projection formulation or numerical approach is used to separate these components in practice?

2.  When generating images from CFG with the effective guidance scale, do the authors use the original pre-trained diffusion models for all methods and apply $w^e$ computed individually for each guidance variant?

---

> ### Author Response · Authors · 2025-11-21
>
> We wish to thank the reviewer for the helpful comments. Please find our answers to the comments below.
>
> **W1:** Writing clarity, especially methodological descriptions.
>
> A: Thank you for the comment. We will carefully proofread our manuscript. Specifically, we have reformulated the description of the effective guidance scale to enhance its clarity and precision.
>
> **W2:** Clear explanations on derivations of Eq (5)-(7).
>
> A: Thank you for pointing this out. We have reformulated the corresponding description in the updated manuscript, please refer to Eq (4)-(9) for details. In general, it only includes basic vector orthogonal decomposition, and does not have any assumptions or approximations.
>
> **W3:** Motivation of TDG.
>
> A: Thanks for the comment. TDG achieves improvement on human-preferenced metrics by adding an additional guidance term $\epsilon^{\mathrm{cond}}_t-\epsilon^{\mathrm{weak}}_t$ and balancing the coefficients between it and $\epsilon^{\mathrm{cond}}_t-\epsilon^{\mathrm{uncond}}_t$. But when conducting ablation studies, we discovered that the performance was mainly contributed by the increase of guidance scale. Meanwhile, many other methods, like PAG, SAG also similar to this process. Therefore, we suspect that similar methods may have exploited this evaluation vulnerability to achieve misleading performance improvement. This is also reflected in our GA-Eval results.
>
> **Q1:** Which projection formulation or numerical approach is used to calculate the components of $\Delta\epsilon$.
>
> A: Good question. We only used the orthogonal projection operation of vectors. Please refer to Eq (4)-(9) in updated manuscript for details, as illustrated in **W2**.
>
> **Q2:** When generating images from CFG with the effective guidance scale, do the authors use the original pre-trained diffusion models for all methods and apply $\omega^\mathrm{e}$ computed individually for each guidance variant?
>
> A: In fact, $\omega^\mathrm{e}$ is calculated by the average of $\omega^\mathrm{e}_t={\||\epsilon_t^\parallel\||}/{\||\Delta\epsilon\||}$ at each timestep. That is to say, $\omega^\mathrm{e}$ is different for different methods, or same method with different prompts. When comparing different methods, we consistently use the same pretrained diffusion model. We have stated the corresponding model used in the captions of Tables 1-9.

---

> ### Author Response · Authors · 2025-11-27
>
> Dear Reviewer:
>
> We hope this message finds you well. As the discussion period is nearing its end, we hope that we have addressed all your concerns satisfactorily. If there are any additional points or feedback you'd like us to consider, please let us know. Your insights are invaluable to us, and we're eager to address any remaining issues to improve our work.
>
> Thank you for your time and effort in reviewing our paper.

---

### Author Response · Authors · 2025-11-21

Thank the reviewers once again for their insightful comment and constructive suggestions. We are happy that the reviewers think that our paper timely and meaningful. We hope that our replies have satisfactorily addressed the reviewers’ questions and further strengthened their assessment of our work. The revised content (reformulated deriviation of $\omega^\mathrm{e}$ in Section 3, and more supplementary results in Appendix C, and D) in the manuscript was labeled on **red**. We remain open to discussion should any issues remain.

---

### Meta-Review · Area_Chair_HCEk · 2026-01-04

**Summary:**

This paper works on improving diffusion guidance evaluation protocol. There are several contributions: 1) authors found common human preference models had strong bias toward large guidance scales; 2) authors introduced guidance-aware evaluation framework for fair comparison for guidance methods; 3) authors designed TGD could improve conventional evaluation framework but doesn't work in practice; 4) experimental results verified effectiveness of proposed evaluation methods.

Before rebuttal, this paper got three 6 ratings and one 2 rating.

The strength of this paper given by reviewers are:
1. insightful observation that commonly used human preference models have strong bias. (reviewer JH4a, 2V4Q, 5pnH, 5uJX)
2. proposed GA-Eval is both reasonable and robust. (reviewer 2V4Q, 5uJX)
3. proposed Transcendent Diffusion Guidance (TDG) method help prove the effectiveness of GA-eval. (reviewer 2V4Q)
4. technical execution is rigorous. (Reviewer 5pnH)
5. Experiments are extensive. (Reviewer 5pnH, 5uJX)

The weakness & questions of this paper given by reviewers are:
1. writing could be improved. (reviewer JH4a)
2. Equations (5)–(7) is not clear. (reviewer JH4a)
3. Motivation of Transcendent Diffusion Guidance is not clear. (reviewer JH4a)
4. need user study. (reviewer 2V4Q)
5. recent reward models like HPSv3 and OneIG might could solve the problem. (reviewer 2V4Q)
6. need more details on why HPS v2, ImageReward, etc., are biased. (Reviewer 5pnH)
7. offers limited direction on how to design guidance methods that truly improve generation beyond CFG scaling. (Reviewer 5pnH)
8. recent text-to-image works might need examination. (Reviewer 5pnH)
9. Theoretical ambiguity in ωₑ definition. (Reviewer 5uJX )
10. Limited validation of GA-Eval fairness. (Reviewer 5uJX)
11. TDG lacks principled formulation. (Reviewer 5uJX)
12. Incomplete quantitative analysis of metric bias. (Reviewer 5uJX)
13. Could GA-Eval itself introduce bias. (Reviewer 5uJX)
14. What is the theoretical motivation for the TDG and How sensitive is TDG to masking ratio or weakened prompt settings. (Reviewer 5uJX)
15. need to provide statistical evidence (e.g., correlation or regression) showing that reward models favor high-saturation images? (Reviewer 5uJX)

In the discussion, Reviewer 5pnH mentioned that authors' response partially addressed their concerns, so they decided to keep the current score 6.

AC read authors' paper, reviewers' comments, authors' rebuttal carefully. And found authors addressed almost all other reviewers concerns (details are in the below session), and thus decided to accept this paper.

**Reviewer Concerns:**

weakness 1. authors mentioned they improve the writing of the paper.

weakness 2. authors improved the equation to make it more clear.

weakness 3. authors explained the motivation of TDG.

weakness 4. authors explained why they don't have user study for Gen-Eval.

weakness 5. authors provided new results on HPSv3, UnifiedReward and OneIG-Bench.

weakness 6. authors provided more results.

weakness 7. authors mentioned GA-Eval does provide some rough direction.

weakness 8. authors provided results on UnifiedReward.

weakness 9. authors mentioned they didn't assume the linearity in the decomposition operation.

weakness 10. authors provided reason why they didn't conduct human evaluation. but didn't mentioned much about fairness.

weakness 11. authors provide some additional results.

weakness 12. authors mentioned they do provide some results on this.

weakness 13. reviewers has concerns on "Could GA-Eval itself introduce bias (e.g., penalizing certain visual styles)?". but authors didn't answer this directly.

weakness 14. authors provided motivations and provided new results.

weakness 15. authors provided more results.

**Reviewer Scores:**

Reviewer JH4a might increase their score from 2.

Reviewer 2V4Q might keep or increase their score 6.

Reviewer 5pnH mentioned they will keep their score 6.

Reviewer 5uJX might keep their score 6.

---

### Decision · Program_Chairs · 2026-01-26

Accept (Poster)